# Wealth, health, and happiness: An inverse story of the Easterlin Paradox in China

**Yangjie Wang[1], Lianhua Li[2], Juan Huang[1], Hongjie Qiang[1]\***

1 School of Business, Central South University, Changsha, China, 2 School of Business, Hunan Institute of Engineering, Xiangtan, China

\* hongjie.qiang@csu.edu.cn

## Abstract

One popular explanation for the Easterlin paradox is that income growth over time is usually accompanied by industrialization and pollution, which cause damage to happiness that cannot be reflected by income change. We examine this explanation by exploring the effects of a large-scale environmental regulation program -the "Two Control Zones (TCZ)" Policy- on subjective well-being (SWB) using data from a series of household surveys in China. We find that, the regulation has successfully mitigated air pollution in the implemented area, although at the cost of local income. Overall, the environmental effect dominates the income effect and TCZ policy increases the SWB of affected people. In particular, despite its negative effect on income, by controlling air pollution, the TCZ policy brought a net increase in residential happiness with a money value of ￥59.04 per month in terms of 2009 CNY. This finding supports the environmental explanation of the Easterlin paradox.

## 1 Introduction

Nowadays, there is a growing concern about sustainable development among academics and policymakers, and with well-grounded reasons. The prevailing pattern of economic development has relied heavily on non-renewable fossil fuels. As a result, it has often been accompanied by severe environmental pollution and increased health risks [1–3], economic growth and wealth accumulation may not necessarily lead to proportional increases in happiness [4]. This concern echoes the well-known Easterlin paradox, which states that although happiness varies positively with income at a point in time, it does not rise over time as income continues to grow [5–7]. One prominent explanation for this paradox centers on environmental degradation. The idea is that long-run income growth is typically accompanied by industrialization and pollution, which impose welfare costs not captured by income measures. Thus, the positive effect of wealth accumulation on happiness could be partially or even totally offset by its environmental cost [8]. In this research, we examine the inverse perspective of the Easterlin paradox. If unsustainable growth accompanied by pollution

**Data availability statement:** All data used in this study are publicly available: The China Health and Nutrition Survey (CHNS) data can be downloaded from the official repository at https://dataverse.unc.edu/dataverse/chns The China Household Income Project (CHIP) data is accessible via application at the official platform: https://bs.bnu.edu.cn/zgjmsrfpdcsjk/sjsq/index.html The $SO_2$ emission data is available for download from the EDGAR database at https://edgar.jrc.ec.europa.eu/dataset_ap81.

**Funding:** This study was financially supported by the National Natural Science Foundation of China in the form of grants received by YW (72088101 and 72173139) and HQ (72203239). This study was also financially supported by the Natural Science Foundation of Hunan Province in the form of a grant (2024JJ2079) received by YW. The funders had no role in the study design, data collection and analysis, decision to publish, or preparation of the manuscript.

**Competing interests:** The authors have declared that no competing interests exist.

reduces SWB, a natural question arises: can environmental regulation -despite potentially slowing economic activity- improve happiness? Or, more specifically, can we improve happiness by implementing environmental regulation on economic behaviors? We explore this question by examining the effects of a large-scale environmental regulation program -the "Two Control Zones (TCZ)" Policy- on subjective well-being (SWB), using household survey data from China, the world's largest developing economy.

The environmental explanation of the Easterlin paradox not only provides a theoretical account of this empirical phenomenon, but also has far-reaching policy implications, especially for governments in developing countries and areas. In such contexts, development paths may be constrained by poverty traps, making environmentally friendly strategies appear unattainable [9–11]. At the early stages of development, countries and areas facing poverty traps might have to sacrifice their environmental quality in exchange for technologies and investments essential for escaping the trap and seizing development opportunities. While such sacrifices cannot continue indefinitely, the key question is when a transition should occur. Examining the environmental explanation of the Easterlin paradox helps to reveal the residents' trade-off between development and environment in developing countries and areas and thus provide important insight for local governments there. This explanation provides a natural starting point for considering whether improvements in environmental quality might increase happiness even at the cost of slower economic growth. If environmental problems associated with development have reduced local SWB, this may indicate that a change in the development trajectory is warranted.

Despite increasing social and academic interest, quantitative literature on the association between income, air pollution and SWB remains in its infancy. Accurately estimating the relationships among income, air pollution, and SWB faces several methodological challenges. First, SWB is influenced by various factors, and many of them could be correlated with air pollution. It is impossible to control all these factors, so directly regressing SWB on air pollution could cause endogenous problems. Second, different demographic groups may have varying preferences for air pollution reduction versus income gains, leading to different levels of willingness to pay (WTP) for environmental quality. This limits the generalizability of findings derived from small or context-specific samples. As a result, context-specific studies often yield ambiguous conclusions. For example, some studies document a significant negative relationship between air pollution and SWB. Others, however, find that pollution plays a relatively minor role compared with factors such as spatial location [12,13], socioeconomic status [14,15], and weather conditions [16]. Similarly, the Easterlin paradox is confirmed in China [3,8,17], Germany [18,19], and Japan [20,21], while the opposite is found in Italy [22–24], the United Kingdom [25,26], and Romania [27].

In this research, we investigate the environmental explanation of the Easterlin paradox taking advantage of a quasi-natural experiment provided by a large-scale environmental regulation program -the "Two Control Zones (TCZ)" Policy- in China. The term "Two Control Zones" refers to acid rain control zones and $SO_2$ pollution control zones. This policy was imposed by the Chinese government in 1998, which

entailed the enforcement of stricter regulations aiming at reducing the use of high-sulfur coal and promoting clean energy technologies. Many researchers have explored the role of the TCZ policy in reducing pollution and improving air quality [28–30], and generally found a significantly positive relationship between the policy and the air quality improvement. However, much less is known about whether this regulation slows economic growth in targeted cities. And if so, how will the trade-off between economic benefit and environmental quality be evaluated by residents and thus affect their SWB? This study aims to explore this dimension of the policy's impact.

We estimate people's willingness to pay for air quality as well as the effect of TCZ policy on air pollution and income. Following Levinson, Ambrey, and Zhang et al. [3,31–33], we find that people's willingness to pay (WTP) is ￥109.56 per month (in 2009 CNY) for a reduction of 10,000 tons in $SO_2$ emissions. Combining this with our estimates of the TCZ policy's effects—reducing $SO_2$ emissions in affected cities by approximately 8,920 tons and lowering average monthly individual income by about 5.2%—we conclude that, on average, the TCZ policy still increases the SWB of affected individuals, with an equivalent monetary value of approximately ￥59.04 per month. Our results are consistent with the environmental explanation of the Easterlin paradox, which suggests that as the economy develops, people increasingly value the environmental quality of their surroundings, and the SWB loss from environmental degradation may outweigh the gains from economic development. In this sense, regulating economic activities to improve air quality may be worthwhile. We conduct several robustness checks to validate this conclusion, including: (1) replacing pollution with self-reported health to verify the pathway through which the policy improves health by reducing pollution, thereby increasing SWB; (2) recalculating WTP while accounting for differences between the CHIP and CHNS datasets; and (3) performing parallel trend tests to assess whether pre-existing trends in income and pollution existed prior to the policy shock; and (4) conducting balancing tests to examine whether the TCZ policy is systematically correlated with observable time-varying characteristics (see Appendix B). The validity of our conclusion is confirmed by all these robustness checks.

The rest of the paper proceeds as follows. Section 2 reviews the literature. Section 3 describes the policy background. Section 4 outlines our empirical strategy, and Section 5 presents the empirical results. Section 6 concludes.

## 2 Literature review

Happiness plays a critical role in economic behavior, and higher life satisfaction is associated with enhanced cognitive functioning, competence, and productivity [34]. Moreover, the field of happiness economics has expanded substantially since the 1990s [35]. It is important to recognize that happiness is a complex concept influenced by a range of individual and societal factors. Prior studies have linked happiness to factors such as income, age, education, GDP per capita, welfare institutions, public insurance, and unemployment rates, among others [35–37]. Among these determinants, the role of wealth accumulation has attracted particular attention, as economic growth remains a primary objective for most governments [38]. The traditional view holds that economic growth raises income, and higher disposable income can be translated into greater consumption and thus higher utility [39]. While conceptually appealing, this view faced empirical limitations due to the difficulty of measuring happiness. In fact, income itself is usually used by economists as a proxy for happiness [12]. Since the 1970s, Easterlin and subsequent researchers have shown that happiness can be elicited through survey questions asking individuals to evaluate their quality of life in subjective terms (such measurements are usually called SWB). Diener et al. [40] further demonstrated that SWB measures exhibit high internal consistency, reliability, and validity.

Research based on SWB challenged traditional beliefs about the income-happiness relationship and generated substantial debate. In U.S. data, Easterlin found that although at a point in time happiness varies directly with income both among and within nations, over time, happiness does not trend upward as income continues to grow [5,7]. Subsequently, this phenomenon, known as the Easterlin paradox, is also confirmed in other developed countries [6,18,41], and later, in less developed countries [41–43]. Critics argue that Easterlin's finding may be due to a failure in isolating statistically significant relationships between average levels of happiness and economic growth through time; related empirical evidence

has been documented in several studies [41,44–47]. Although there is debate regarding the significance of the effect, both supporters and opponents of the Easterlin paradox generally report positive but very small estimates of the stated long-run income-happiness relationship [48]. It seems that even if income does have a long-run influence on happiness, the relationship is not as prominent as traditional beliefs.

Several theories attempt to explain why the long-run relationship between income and happiness appears weak. Easterlin and his followers argue that this phenomenon might be due to psychological issues of hedonic adaptation [49–51], or maybe it is the relative position of the income instead of its absolute value which determines happiness [47,50,52–54]. Beyond psychological explanations, other researchers seek to explain the Easterlin paradox from an economic perspective. They suggest that happiness is neither a definite nor an automatic consequence of income growth [48]; instead, it also consists of many other things such as interpersonal relationships [55], social security [56], and health [57]. One popular explanation from the economic perspective concerns the environment. This explanation suggests that income growth over time is usually accompanied by industrialization and pollution, which harm happiness in ways that are not captured by changes in income. As a result, pollution accompanying income growth may offset the positive effects of wealth accumulation on SWB [8]. In recent years, as concerns about environmental quality, sustainable development, and climate change have increased, policymakers and academics have shown increasing interest in this explanation of the Easterlin paradox and have used it to justify environmental regulations that are often criticized for potentially harming the economy and employment [58,59].

Two main lines of research have focused on the environmental explanation of the Easterlin paradox. First, abundant literature has examined the relationship between pollution and SWB [3,60–62]. These studies examine multiple types of pollution and generally show that pollution associated with economic growth can reduce SWB. However, most of these studies examine how pollution affects happiness while treating income as exogenous. Few have considered the endogenous relationship between income and pollution, or explored whether environmental regulation can improve SWB by accounting for both effects. By estimating the welfare effects of the TCZ policy, our study provides an integrated perspective on both the income and environmental channels. A recent study comparable to ours in this regard is Sun et al. [63], which considers both income and environmental effects as transmission mechanisms through which air pollution affects SWB. However, Sun et al. [63] use provincial-level cross-sectional data in their analysis, which makes their results more prone to endogeneity issues. All cross-city variations within the province are also averaged out in their research. By contrast, we construct a panel dataset based on a longitudinal survey and match interviewees' air pollution exposure and SWB at the city level.

Second, in recent years, empirical studies based on hedonic pricing or contingent valuation have emerged to estimate how an individual's SWB is affected by amenities (including pollution) in their living surroundings, driven by the growing need to evaluate people's WTP for environmental policies or business projects. [64–67]. Studies using hedonic pricing or contingent valuation provide important insights into how people make trade-offs between wealth and the environment in specific contexts. However, due to the high requirement for detailed individual information, such research usually either focuses on a small group of interviewees concentrated in limited geographic areas (see Borja-Urbano et al. [67]), or relies on aggregated data at the provincial or national level (see Malpezzi [68]). They cannot provide a comprehensive view of how regional or local happiness profiles are affected by the corresponding environmental conditions. We use pooled cross-sectional data covering 214 cities across 22 provinces from 2002 to 2013, matching city-level pollution measures and economic variables to individual respondents. This structure allows us to capture more comprehensively how environmental conditions affect SWB. Our research can also be linked to the growing environmental regulation literature on China, the world's largest developing economy. In recent years, the Chinese government has introduced a series of environmental regulations to address the growing threat of environmental deterioration resulting from rapid industrialization and economic growth. Existing literature has examined the effectiveness of various environmental regulations in China from various perspectives [62,69–71]. Extensive research has also examined the effects of the TCZ policy. However,

due to data limitations, most of the existing literature has to focus on only one aspect of the policy's effects. For example, several studies examine the pollution reduction or health effect of the policy [30,72–74], while others focus on its impact on income [75,76].

Our study contributes to the environmental regulation literature and the TCZ policy literature by: 1) comprehensively examining two key mechanisms—pollution and health effects, and income effects—through which the TCZ policy influences SWB in the context of the Easterlin paradox; and 2) quantifying the monetary value of the TCZ policy's impact using a WTP-based approach.

## 3 Background

To contextualize our empirical strategy, we begin with a brief overview of China's air pollution challenges and its efforts to curb $SO_2$ emissions, with a particular emphasis on the TCZ policy.

China has experienced remarkable economic growth since the launch of its reform and opening-up policy and its transition toward a market economy in the late 1970s. However, as in many developing countries, rapid industrialization brought not only substantial economic growth but also severe environmental degradation. As early as the 1990s, coal combustion–induced $SO_2$ emissions represented one of the most serious air pollution problems in China [77]. Elevated atmospheric $SO_2$ concentrations pose serious health risks, increasing the likelihood of respiratory diseases such as lung cancer and causing irritation of the conjunctiva and upper respiratory tract, among other inflammatory conditions [78] $SO_2$ emissions are also a major contributor to acid rain. In 1995, approximately 40% of the country's territory reported acid rain with an average pH value below 5.6 [79]. There were growing concerns that acid rain could damage ecosystems, buildings, and human health in China [80,81].

The Chinese government recognized the threats of $SO_2$ emissions and acid rain and adopted a series of measures to address this problem beginning in the 1980s [82]. China's early efforts to combat $SO_2$ emissions and acid rain included the Air Pollution Prevention and Control Law (APPCL) enacted in 1987, its amendment in 1995, and a series of national and local emission standards gradually implemented since 1990. However, the effectiveness of these tentative and unsystematic efforts was questioned by many researchers [83–85]. Therefore, in 1998, the Chinese government launched a systematic regulatory framework for $SO_2$ emissions, namely the "Two Control Zones (TCZ)" policy. Specifically, 175 cities across 27 provinces were designated as either acid rain control zones (mainly in South China) or $SO_2$ pollution control zones (mainly in North China), and were subjected to a series of strict air pollution regulations. The TCZs covered approximately 1.09 million square kilometers, accounting for 11.4% of China's territory, 40.6% of the population and 58.9% of the total $SO_2$ emissions in 1995. Fig 1 shows the geographic distribution of TCZ cities in China.

Unlike previous efforts, this ambitious policy initiative fully reveals China's determination to solve $SO_2$ emission-related problems. First, the TCZ policy established detailed production requirements for related industries. For example, according to the *Official Reply of the State Council on Issues Concerning Acid Rain Control Areas and Sulfur Dioxide Pollution Control Areas* issued in 1998, new coal mines in TCZ cities can obtain approval only if the sulfur content in their products is controlled below 3%. Existing coal mines that fail to meet this requirement are required to reduce their production or even suspend operations until compliance is achieved. As a major emission source, coal-combustion power plants face even stricter requirements. Newly constructed plants burning coal with sulfur content higher than 1.5% must install sulfur scrubbers before being approved for operation. Existing plants burning coal with similar sulfur content must adopt effective $SO_2$ emissions reduction measures by 2000. Moreover, new plants were no longer allowed to be built near large cities, except for heat-supplying cogeneration purposes. Production procedures in industries with $SO_2$ emissions potential such as steel, cement, electric power, nonferrous metals, etc., are also strictly regulated. These detailed requirements make the TCZ policy not only a principal guide for $SO_2$ emission reduction but also an enforceable operational manual for both central and local governments.

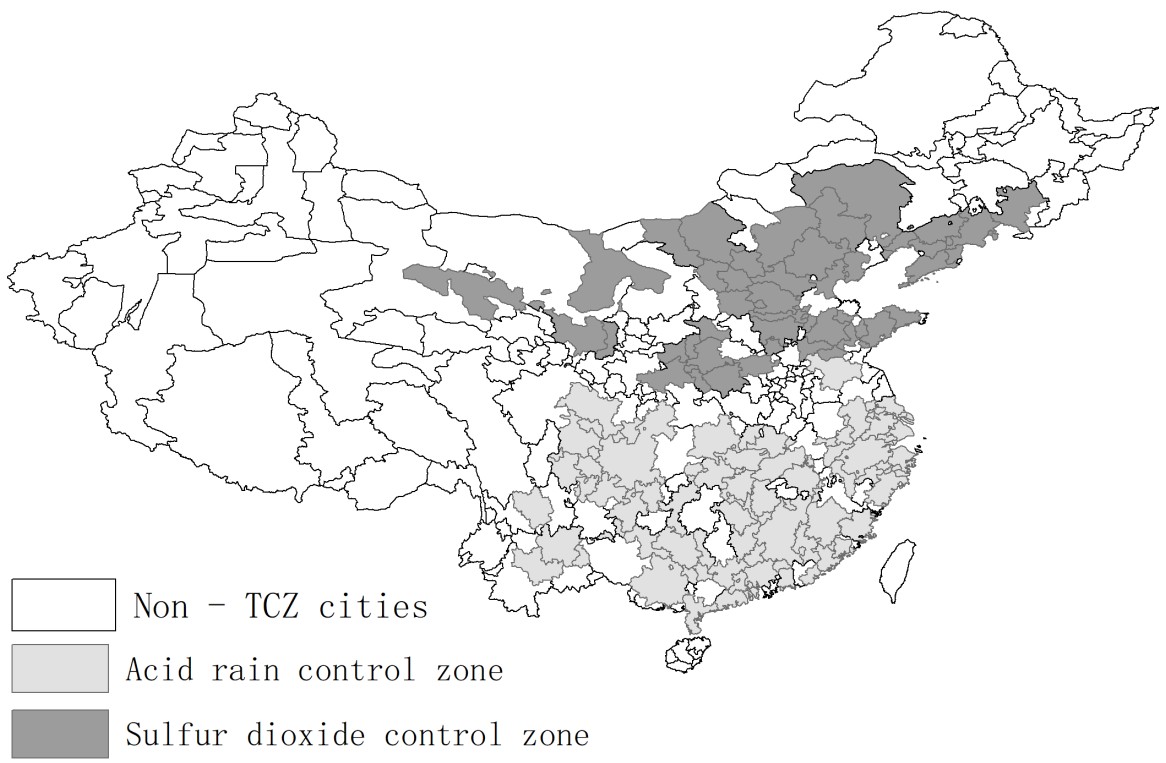

**Fig 1. Geographic distribution of TCZ cities. Notes:** Map created by the authors using administrative boundary data from GeoBoundaries [86]. The southern Chinese archipelago is not shown in the figure.

Second, the TCZ policy profile was executed with strong administrative and fiscal support. Administratively, local governments and $SO_2$-emitting industries—especially power plants and coal producers—were required to adopt effective measures in accordance with TCZ plans to ensure the achievement of emission reduction goals. The Environmental Protection Agency, the Ministry of Economic and Trade, and the Ministry of Science and Technology were designated to guide $SO_2$ emissions reduction projects, monitor atmospheric $SO_2$ concentrations, and inspect emission control efforts. Fiscally, China expanded its $SO_2$ discharge fee system, piloted in selected areas since 1992, to cover the entire TCZ region. The collected funds were used as local environmental protection subsidies, with over 90% allocated to $SO_2$ control projects targeting key polluters. During the *Tenth Five-Year Plan* period (2001~2005), the central government also earmarked 12 billion CNY for TCZ cities to construct desulfurization facilities in coal-fired power plants, aiming to reduce annual $SO_2$ emissions by 1.05 million tons.

Third, to ensure the effectiveness of the policy, the State Council set both short-term (by 2000) and long-term (by 2010) pollutant emission control targets for each TCZ city. Most cities ultimately met their assigned targets. For example, according to Shenyang City's *Eleventh Five-Year Major Pollutant Emission Reduction Work Plan*, the city aimed to reduce its $SO_2$ emissions to below 105.4 thousand tons per year by 2010; the actual emission level was 96.4 thousand tons. Similarly, Shanghai City's *Eleventh Five-Year Plan for environmental protection and ecological construction* set a target of keeping annual $SO_2$ emissions below 380 thousand tons by the end of 2010; the actual level reached 358.1 thousand tons.

With detailed regulatory measures, strong fiscal and administrative backing, and clear pollution control targets integrated into government performance evaluations, the TCZ policy marked a milestone in China's efforts to combat

$SO_2$-related environmental problems. A large body of official documentation has confirmed the TCZ policy's effectiveness in improving air quality. By 2005, 45.1% of cities in the $SO_2$ pollution-control zones met the national Class II standard for average ambient $SO_2$ concentrations, while 73.9% of cities in the acid rain-control zones had done so. By 2010, the proportion of TCZ cities meeting the national Class II standard had risen to 94.9%.

Theoretically, the TCZ policy can influence residents' SWB via two major channels. The first is the well-documented pollution–health channel. As suggested by existing literature, an increase in $SO_2$ concentration in the atmosphere can cause chronic obstructive pulmonary disease [87], asthma [88,89], bronchiectasis [90] and tuberculosis [81]. All these deteriorate the health status or self-reported health status of affected residents and lower their happiness, as numerous studies have found a strong link between health and SWB across different age groups [37,91–93]. In this sense, TCZ policy could increase happiness by improving self-recognized health status. The second is the more contentious income channel. On the one hand, pollution control regulations compel firms to alter production inputs or invest in abatement technologies, thereby raising costs and reducing output, profitability, and ultimately employment and wages [94]. From this perspective, TCZ policy may adversely impact residents' income. On the other hand, sustained regulations may incentivize firms to pursue long-term green innovations, thereby enhancing productivity and potentially increasing employment and wages [95,96]. Better health and environmental quality have also been shown to boost labor productivity, increase labor supply, and raise total earnings [97–99]. Moreover, environmental governance itself can generate employment [94]. Thus, the TCZ policy may also increase average income. Fig 2 presents the two mechanisms through which TCZ policy could potentially influence SWB. In summary, we expect the TCZ policy to reduce $SO_2$ emissions and improve perceived health among residents. However, whether it reduces or increases income—and which channel has a greater effect on SWB—remains an open question in our study. As a preview of our findings, the TCZ policy significantly reduced $SO_2$ emissions and improved residents' perceived health, albeit at the cost of individual income and household income. Overall, the health benefits outweighed the income losses, resulting in increased SWB. This provides support for the environmental explanation of the Easterlin paradox in China.

## 4 Empirical strategy

This study investigates the environmental explanation of the Easterlin paradox, which posits that beyond a certain income threshold, increases in SWB may be offset by pollution generated during economic growth, thereby weakening or even eliminating the observed correlation between income and SWB. We examine this question from an inverse perspective by analyzing how an air pollution regulation that successfully reduced targeted emissions affected individuals' SWB. If the environmental explanation of the Easterlin paradox holds, we should observe that effective air pollution regulations can improve SWB despite potential negative income effects.

### 4.1 Data

To address our research questions, we combine data from the following sources.

Our first data source comprises two large-scale household surveys: the China Health and Nutrition Survey (CHNS) and the China Household Income Project (CHIP).

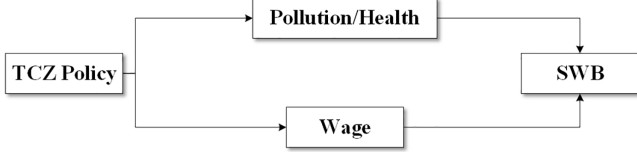

**Fig 2. Two mechanisms TCZ policy affects SWB.**

The CHNS provides a longitudinal panel dataset derived from individual and community-level surveys. It was jointly conducted by the Carolina Population Center at the University of North Carolina at Chapel Hill and the National Institute of Nutrition and Food Safety at the Chinese Center for Disease Control and Prevention, with the aim of investigating nutrition and health among Chinese residents. As of 2015, ten waves of CHNS surveys had been conducted, covering both urban and rural areas in 15 regions across China, including 12 provinces and 3 municipalities (Beijing, Chongqing, Shanghai, Guangxi, Guizhou, Heilongjiang, Henan, Hubei, Hunan, Jiangsu, Liaoning, Shandong, Shanxi, Yunnan, and Zhejiang). For this analysis, we use data from six waves of the CHNS conducted between 1991 and 2006 (1991, 1993, 1997, 2000, 2004, and 2006), which provide two key variables: self-reported health status (SRH) and individual income. Respondents were asked, "How would you describe your health compared to that of other people at your age?" The responses are categorized into three categories: good, fair, and poor. In addition, respondents reported their income from their primary job.

The CHNS provides extensive information relevant to this study, including individual characteristics (such as SRH, age, sex, education, income, and marriage), household information (such as household size, number of kids, and household income), and community-level information (such as community medical service level, community education level and community hygiene level). The multistage, random cluster process used to draw the sample, along with the longitudinal nature of the data, also allows us to better control for unobserved individual heterogeneity and potential selection bias. The CHNS sample used in our analysis includes approximately 11,088 observations, covering 9 provinces and 47 cities (21 of which are TCZ cities and 26 are non-TCZ cities). It involves 2,351 households and 3,886 individuals, with data spanning from 1991 to 2006. However, the CHNS has one notable limitation. The survey focuses on the health status of the residents and does not include questions about subjective wellbeing. Therefore, we incorporate the CHIP survey to examine how income, pollution, and consequently health affect SWB—the right-hand side of Fig 2.

The CHIP surveys are conducted by the Agricultural Survey Corps in the National Bureau of Statistics of China and the Institute of Economics in the Chinese Academy of Social Sciences using methodologies closely aligned with those of the NBS surveys (see Eichen and Ming [100]; Li et al. [101]; Luo and Shi [102]). Currently, the surveys have been conducted in five rounds across most regions of China and the questionnaires cover topics such as income, consumption, employment, and production. Since 2003, the surveys have included the question: "Do you feel happy now?" The options include excellent, good, fair, poor, very poor, and unknown. The CHIP dataset has the advantage of containing information on both SRH and SWB; however, these variables were introduced only after the implementation of the TCZ policies. In this research, we utilize both surveys to achieve our research objectives. The CHIP sample used in our analysis includes 23,187 observations, covering most regions across the country, and the data is from 2002, 2007, 2008, and 2013.

The second data source used in this study is the city-level annual $SO_2$ emission data, obtained from the Emissions Database for Global Atmospheric Research (EDGAR v8.0) [103]. EDGAR provides harmonized and internationally comparable estimates of anthropogenic emission inventories. Based on this dataset, we construct annual $SO_2$ emissions for each city from 1991 to 2013 to measure air pollution intensity. We use emission data rather than ambient concentration data because the TCZ policy primarily targeted industrial $SO_2$ emissions.

The third dataset includes information on city characteristics, including weather conditions, obtained from the National Meteorological Information Center. Table 1 summarizes descriptive statistics for each variable by data source.

## 4.2 Estimation

The main empirical challenge of this study is the lack of consistent SWB data before and after the implementation of the TCZ policy. This issue is common in studies on SWB, as obtaining reliable and comparable SWB measures is inherently difficult. Fortunately, we have access to two complementary datasets that allow us to combine their strengths and offset

**Table 1. Summary statistics.**

| | | CHNS | | CHIP | |
|---|---|---|---|---|---|
| Variable | Definition | Mean | SD | Mean | SD |
| SWB | Very unhappy = 1; unhappy = 2; fair = 3; happy = 4; very happy = 5 | – | – | 3.930 | 0.830 |
| Income | Individual monthly income (CNY) | 743.9 | 955.2 | 1954 | 2224 |
| Health | Unhealthy = 1; fair = 2; healthy = 3 | 2.759 | 0.468 | 2.720 | 0.510 |
| Age | Age (years; individuals older than 16) | 9.695 | 3.299 | 10.31 | 3.390 |
| Education | Years of formal education | 39.13 | 10.34 | 43.26 | 9.770 |
| Marriage | Never married = 1; married = 2; divorce = 3; widowed = 4 | 1.906 | 0.399 | 2 | 0.350 |
| Household income | Annual household income (1,000 CNY) | 26.45 | 26.86 | 47.85 | 48.10 |
| Household members | Household size (number of members) | 3.914 | 1.399 | 3.360 | 1.070 |
| $SO_2$ | Annual average $SO_2$ emissions (in ten thousand tons) | 9.978 | 9.915 | 16.95 | 12.42 |
| Rain | Annual average precipitation (mm) | 1056 | 436.3 | 1080 | 493.4 |
| Temperature | Annual average temperature (°C) | 14.43 | 4.590 | 16.13 | 3.620 |
| Wind speed | Annual average wind (m/s) | 2.430 | 0.801 | 2.150 | 0.520 |

their respective limitations. As noted in Section 4.1, the CHNS is a ten-wave longitudinal survey conducted from 1989 to 2015, covering periods both before and after the 1998 TCZ policy. It includes rich individual-, household-, and community-level information, with the exception of SWB. If SWB were also included, the CHNS would be the ideal dataset for our research. In contrast, beginning in 2002, the CHIP survey started collecting relatively complete responses to questions on SWB. There have been three survey rounds since then, each providing valuable information such as household income, wages, and self-reported health—essential variables for predicting how living conditions influence SWB. Among these, income and health are of particular interest, as they represent the two main channels through which the TCZ policy could affect SWB. Thus, the CHNS allows us to estimate how the TCZ policy affects pollution and income (the left half of Fig 2), whereas the CHIP enables us to identify how income, pollution (or health), and SWB are jointly related (the right half of Fig 2). Combining the two datasets allows us to address our central question: Can air pollution regulation improve happiness? The idea is as follows:

First, we use a typical difference-in-difference (DID) approach to estimate the effect of the TCZ policy on air pollution and residents' income. For air pollution, we regress:

$$P_{jpt} = \alpha_1 + \beta_1 \left( TCZ_j \times Post_t \right) + \omega \left( P \times f(t) \right) + \varphi_1 W_{jpt} + \delta_j + \theta_t + \mu_{pt} + \varepsilon_{jt} \tag{1}$$

Here, $P_{jpt}$ is the air pollution of city $j$ (located in province $p$) at time $t$, measured by the annual $SO_2$ emissions (in ten thousand tons) at the city level. $TCZ_j$ is a dummy variable equal to 1 if city $j$ is subject to the TCZ policy. $Post_t$ is a dummy variable equal to 1 for years 1998 and thereafter, representing the post-policy period. $\hat{\beta}_1$ is then the parameter of interest which illustrates the effect of TCZ policy on city air pollution. To address the non-random assignment of the TCZ policy, following Li et al. [104] and Liu et al. [105], we additionally include the interaction term $P \times f(t)$, where $P$ represents the baseline $SO_2$ emissions of each city in 1997(the pre-policy year), and $f(t)$ is specified as (i) year fixed effects, (ii) a post-policy dummy, or (iii) a third-order polynomial in time. These specifications flexibly control for differential pre-policy trends associated with initial pollution levels. For comparison, we also report the corresponding regression results and parallel-trend tests from specifications that exclude the interaction term $P \times f(t)$ in Appendix A (Tables A1-A2 and Figures A1-A2). The regression further controls for city-level meteorological variables such as precipitation, temperature, and wind speed as well as the city fixed effect $\delta_j$, the time fixed effect $\theta_t$, and the combined fixed effect at a higher dimension, i.e., $\mu_{pt} = Province_p \times time_t$. $\varepsilon_{jpt}$ is a heteroskedasticity-robust standard error term.

For residential income, we regress:

$$InR_{ihjpt} = \alpha_2 + \beta_2 \left( TCZ_j \times Post_t \right) + \eta_2 X_{ihjpt} + \phi_2 M_{hjpt} + \varphi_2 W_{jpt} + \omega \left( P \times f(t) \right) + \tau_i + \delta_j + \theta_t + \mu_{pt} + \varepsilon_{ihjpt} \tag{2}$$

Here, $InR_{ihjpt}$ is the log monthly income of individual $i$ from household $h$ living in city $j$, province $p$, at time $t$. $X_{ihjpt}$ is a series of time-varying individual level control variables including years of education, age, and marital status. $M_{hjpt}$ is a series of household level control variables including the household's income and family size. $\tau_i$ is the individual fixed effects. $W_{jpt}$, $\left( P \times f(t) \right)$, $\delta_j$, $\theta_t$ and $\mu_{pt}$ are the same as in equation (1), and $\hat{\beta}_2$ is the parameter of interest which measures the percentage change in monthly income associated with the TCZ policy. The above two regressions are estimated using the CHNS data.

Second, following Levinson, Ambrey, and Zhang et al. [3,31–33], we estimate residents' WTP for improved air quality by exploring their trade-offs between economic benefit and environmental quality subject to the constraint that their SWB stay the same. Since income might be correlated with unobserved factors that also affect SWB, we use an instrumental-variable (IV) approach to address potential endogeneity.

In terms of individual income, we choose the provincial level average income of workers who work in the same industry as the instrumental variable for individual income, as suggested by Zhang et al. [3]. We use a two-stage least squares (2SLS) approach to estimate this IV model. The 1st stage regressions are:

$$InR_{ihjkpt} = \alpha_3 + \beta_3 InAvgWage_{kpt} + \eta_3 X_{ihjpt} + \phi_3 M_{hjpt} + \varphi_3 W_{jpt} + \delta_j + \theta_t + \mu_{pt} + \varepsilon_{ihjpt} \tag{3}$$

In equation (3), $InR_{ihjkpt}$ is the log monthly income of individual $i$ at time $t$, where individual $i$ is from household $h$, lives in the city $j$ of province $p$, and works in industry $k$. $InAvgWage_{kpt}$ is the average log income for staff who work in the same industry and the same province as individual $i$. Control variables $X_{ihjpt}$, $M_{hjpt}$, $W_{jpt}$, fixed effects $\delta_j$, $\theta_t$ and $\mu_{pt}$ are the same as in equation (1) and (2).

In the 2nd stage, we regress the following:

$$SWB_{ihjkpt} = \alpha_4 + \beta_4 P_{jpt} + \gamma_4 In\hat{R}_{ihjkpt} + \eta_4 X_{ihjpt} + \phi_4 M_{hjpt} + \varphi_4 W_{jpt} + \delta_j + \theta_t + \mu_{pt} + \varepsilon_{ihjpt} \tag{4}$$

Here, $In\hat{R}_{ihjkpt}$ is predicted from equation (3). In equation (4), the parameters of interest are $\hat{\beta}_4$ and $\hat{\gamma}_4$, as they determine the change in SWB while holding other factors constant:

$$\Delta \widehat{SWB}_{ihjkpt} = \hat{\beta}_4 \Delta P_{jpt} + \hat{\gamma}_4 \Delta InR_{ihjkpt} \tag{5}$$

Therefore, according to Levinson, Ambrey, and Zhang et al. [3,31–33], the WTP of affected residents for improved air quality (measured by reduced $SO_2$ emissions) can be calculated by:

$$\widehat{WTP} = \left. \frac{\partial R}{\partial P} \right|_{dSWB = 0} = -\overline{R}_{CHIP} \frac{\hat{\beta}_4}{\hat{\gamma}_4} \tag{6}$$

Here, $\overline{R}_{CHIP}$ is the average income of the sample. Equation (3) to (6) are estimated using the CHIP data.

Finally, with the influence of TCZ policy on air pollution ($\hat{\beta}_1$) and on the percentage change in monthly income ($\hat{\beta}_2$) estimated in the first step, together with residents' willingness to pay (WTP) for reduced air pollution obtained from the second step, we calculate the net monetary value of the TCZ policy for affected residents as:

$$Value(\Delta SWB) = \hat{\beta}_1 \widehat{WTP} + \hat{\beta}_2 \overline{R}_{CHNS} \tag{7}$$

## 5 Results

### 5.1 Main results

This section presents the main results on how the TCZ policy affects the subjective well-being (SWB) of affected residents. As previously described, the analysis combines information from two large-scale household surveys: CHNS and CHIP.

We first apply the DID approach to the CHNS data and find that the TCZ policy significantly improves air quality in affected cities, although it comes at the cost of reduced residents' income. Table 2 reports the estimated effect of the TCZ policy on air quality, measured by city-level $SO_2$ emissions.

As shown in Column (1) of the table, we find a significant negative effect of the TCZ policy on city-level $SO_2$ emissions at the 1% level. More specifically, the implementation of the policy caused $SO_2$ emissions in affected cities decrease by about 8,920 tons. This finding is consistent with Huang et al. [72], who also document a decline in $SO_2$ emissions following the TCZ policy.

Table 3 reports the estimated economic effects of the TCZ policy, using the log of monthly income of surveyed individuals as the outcome variable. As shown in Column (1), after controlling for a set of observable characteristics that may affect individual income, the TCZ policy has a significant negative effect on residents' monthly earnings. Specifically, the implementation of the TCZ policy reduces monthly income by approximately 5.2%, equivalent to a loss of about ￥38.68 relative to the sample average monthly income of ￥743.9. This estimate is statistically significant at the 1% level. This finding is consistent with Sun et al. [75], who also document income reductions in small and medium-sized cities following the TCZ policy.

**Table 2. The impact of the TCZ Policy on $SO_2$.**

| VARIABLES | (1) | (2) | (3) |
|---|---|---|---|
| | $SO_2$ | $SO_2$ | $SO_2$ |
| TCZ×Post | −0.892*** | −0.800*** | −0.827*** |
| | (0.312) | (0.295) | (0.283) |
| Rain | −0.000 | −0.001 | −0.000 |
| | (0.000) | (0.000) | (0.000) |
| Temperature | −0.719** | −0.737** | −0.721** |
| | (0.309) | (0.367) | (0.310) |
| Wind speed | −0.918 | −1.416** | −1.001 |
| | (0.586) | (0.643) | (0.633) |
| P×Year dummy | YES | NO | NO |
| P×Post$_{1998}$ | NO | YES | NO |
| P×t | NO | NO | YES |
| P×t×t | NO | NO | YES |
| P×t×t×t | NO | NO | YES |
| Year FE | YES | YES | YES |
| City FE | YES | YES | YES |
| Province-by-year FE | YES | YES | YES |
| Observations | 745 | 745 | 745 |
| R-squared | 0.961 | 0.947 | 0.960 |

Note: Table 2 presents the estimation results of Equation (1). Columns (1)–(3) correspond to different specifications of the time function $f(t)$. Column (1) uses time fixed effects; Column (2) employs a post-policy dummy; and Column (3) adopts a third-order time polynomial. Robust standard errors are reported in parentheses (*** p<0.01, ** p<0.05, * p<0.1).

**Table 3. Impact of the TCZ Policy on log monthly income.**

| VARIABLES | (1) Ln(R) | (2) Ln(R) | (3) Ln(R) |
|---|---|---|---|
| TCZ×Post | −0.052** | −0.050** | −0.044* |
|  | (0.025) | (0.025) | (0.024) |
| Education | 0.006 | 0.006 | 0.006 |
|  | (0.006) | (0.006) | (0.006) |
| Age | 0.107 | 0.100 | 0.107 |
|  | (0.070) | (0.070) | (0.070) |
| Age² | −0.050*** | −0.050*** | −0.049*** |
|  | (0.007) | (0.007) | (0.007) |
| Married | 0.047* | 0.046* | 0.047* |
|  | (0.026) | (0.026) | (0.026) |
| Divorce | 0.153*** | 0.153*** | 0.154*** |
|  | (0.059) | (0.059) | (0.059) |
| Widowed | 0.093 | 0.097 | 0.096 |
|  | (0.080) | (0.081) | (0.081) |
| Household income | 0.008*** | 0.008*** | 0.008*** |
|  | (0.001) | (0.001) | (0.001) |
| Household members | −0.026*** | −0.026*** | −0.026*** |
|  | (0.008) | (0.008) | (0.008) |
| Rain | 0.000 | 0.000* | 0.000* |
|  | (0.000) | (0.000) | (0.000) |
| Temperature | 0.036 | 0.032 | 0.033 |
|  | (0.024) | (0.024) | (0.024) |
| Wind speed | 0.006 | −0.001 | 0.000 |
|  | (0.038) | (0.036) | (0.038) |
|  | (85.729) | (80.392) | (88.977) |
| P×Year dummy | YES | NO | NO |
| P×Post$_{1998}$ | NO | YES | NO |
| P×t | NO | NO | YES |
| P×t×t | NO | NO | YES |
| P×t×t×t | NO | NO | YES |
| Individual FE | YES | YES | YES |
| Year FE | YES | YES | YES |
| City FE | YES | YES | YES |
| Province-by-year FE | YES | YES | YES |
| Observations | 11,088 | 11,088 | 11,088 |
| R-squared | 0.799 | 0.799 | 0.799 |

Note: Table 3 presents the estimation results of Equation (2). Columns (1)–(3) correspond to different specifications of the time function $f(t)$. Column (1) uses time fixed effects; Column (2) employs a post-policy dummy; and Column (3) adopts a third-order time polynomial. Robust standard errors are reported in parentheses (*** $p < 0.01$, ** $p < 0.05$, * $p < 0.1$).

We then use the CHIP data to estimate the relationship between income, air quality and SWB using a 2SLS IV regression. We report the related results in Table 4, where Columns (1) represents the 1st stage results while Column (2) represents the 2nd stage results. In Column (1), the instrument (Ln(Avgwage)) has a strong and statistically significant effect

**Table 4. The relationship between income, air quality and SWB.**

| VARIABLES | (1)<br>First stage<br>Ln(R) | (2)<br>Second stage<br>SWB |
|---|---|---|
| Ln(Avgwage) | 0.339*** | |
| | (0.029) | |
| $SO_2$ | 0.002 | −0.006* |
| | (0.004) | (0.003) |
| Ln(R) | | 0.107* |
| | | (0.059) |
| Education | 0.058*** | 0.012*** |
| | (0.003) | (0.004) |
| Age | 0.056*** | −0.036*** |
| | (0.005) | (0.005) |
| Age^2 | −0.071*** | 0.040*** |
| | (0.006) | (0.006) |
| Married | 0.041 | 0.208*** |
| | (0.031) | (0.028) |
| Divorce | 0.084* | −0.278*** |
| | (0.043) | (0.053) |
| Widowed | −0.070 | −0.286*** |
| | (0.052) | (0.066) |
| Household income | 0.000*** | 0.000* |
| | (0.000) | (0.000) |
| Household members | −0.080*** | 0.021*** |
| | (0.008) | (0.007) |
| Rain | 0.000*** | −0.000 |
| | (0.000) | (0.000) |
| Temperature | 0.013 | −0.083 |
| | (0.055) | (0.056) |
| Wind speed | 0.064 | −0.084 |
| | (0.068) | (0.058) |
| Year FE | YES | YES |
| City FE | YES | YES |
| Province-by-year FE | YES | YES |
| Cragg-Donald Wald F-statistic | 165.782 | |
| Observations | 23,187 | 23,187 |
| R-squared | 0.366 | 0.030 |

Note: Table 4 reports the estimation results of Equation (3) and Equation (4). Column (1) presents the first-stage IV regression corresponding to Equation (3). The Cragg-Donald Wald F statistic (165.782) and the Kleibergen-Paap rk Wald F statistic (140.884) both exceed conventional thresholds, indicating that the instrument is strongly identified. Column (2) reports the second-stage IV estimates corresponding to Equation (4). Robust standard errors are shown in parentheses. (*** $p < 0.01$, ** $p < 0.05$, * $p < 0.1$).

on the endogenous variable (Ln(R)). The reported F-statistics exceed conventional thresholds for weak identification, indicating that weak instruments are unlikely to be a concern in our specification. In Column (2), we obtain $\hat{\beta}_5$ = –0.006 and $\hat{\gamma}_5$ = 0.107. Since we use a linear 2SLS model in this regression, these two estimates do not have direct economic

interpretations. However, their ratio reveals the trade-off between income and air pollution required to keep the individual's perceived level of subjective well-being (e.g., feeling excellent, good, fair, poor, or very poor) unchanged. Together with the average income of the sample, i.e., ￥1954 per month, we estimate that the average monthly willingness to pay for a 10,000-ton reduction in $SO_2$ emissions is ￥109.56.

Finally, we combine the DID and IV results to estimate the monetary value of the net SWB improvement generated by the TCZ policy for affected residents. This value is calculated as:

$$Value(\Delta SWB) = \left( \hat{\beta}_1 \widehat{WTP} + \hat{\beta}_2 \overline{R}_{CHNS} \right) = 0.892 \times 109.56 - 0.052 \times 743.9 = 59.04.$$

### 5.2 Robustness checks

In this section, we present a series of robustness checks to assess the validity of our main results.

First, a standard concern in DID analyses is whether treated and control groups follow parallel trends prior to the policy shock. We therefore conduct two parallel-trends (event-study) tests.

Fig 3 displays the event-study estimates for Equation (1), which test the parallel-trends assumption for the TCZ effect on city $SO_2$ emissions. The coefficients prior to the policy implementation are statistically indistinguishable from zero, while post-policy coefficients become negative. These results support the parallel-trends assumption underlying our DID design.

Fig 4 reports the event-study estimates for Equation (2). The pre-policy coefficients are not statistically significant, indicating no differential trends in log monthly income between treatment and control groups prior to implementation. This pattern supports the validity of the parallel-trends assumption for the income specification.

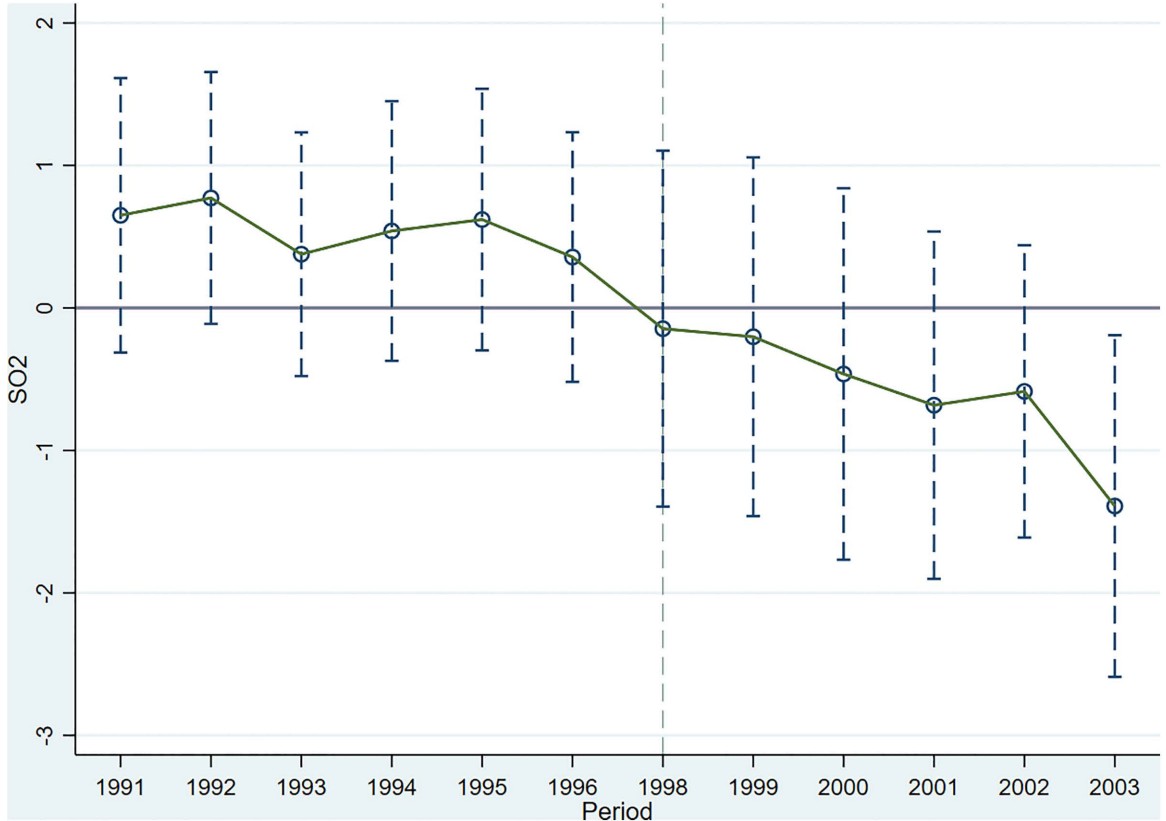

**Fig 3. Parallel Trend Test (the TCZ on SO₂).** Note: The baseline year is 1997, and the policy implementation year is 1998.

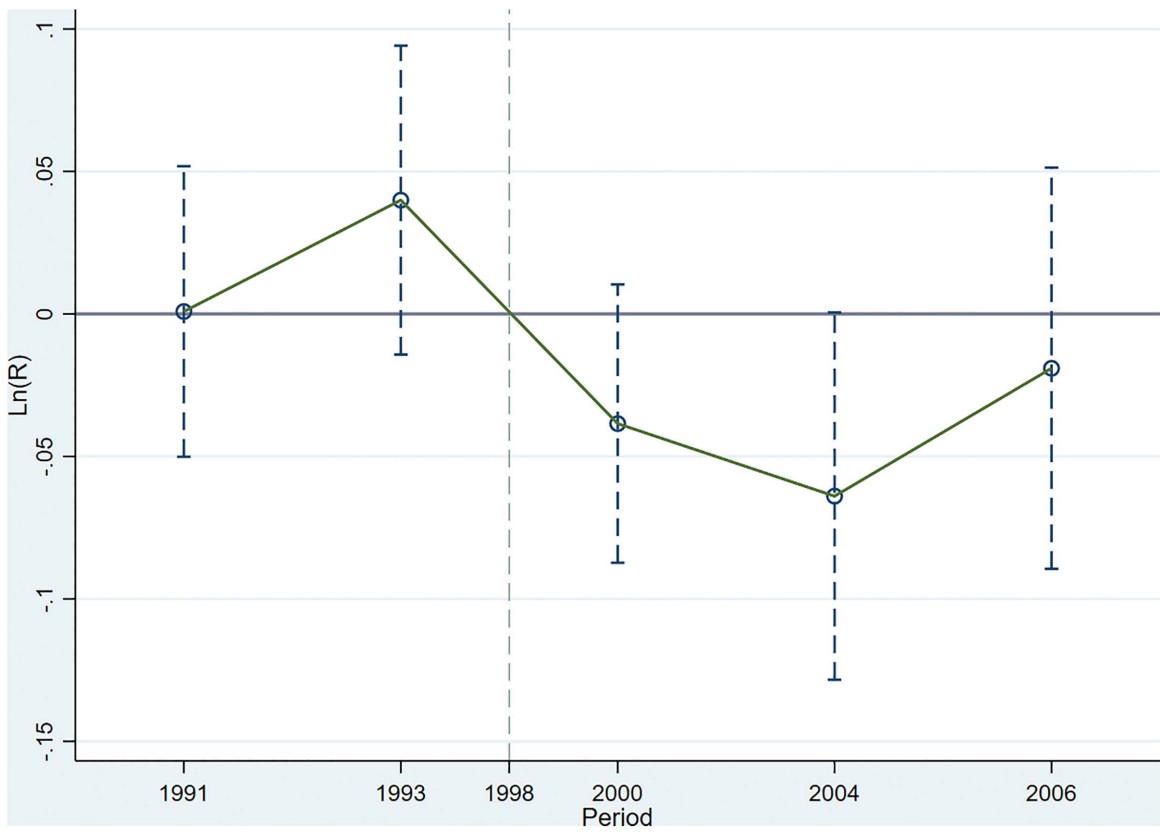

**Fig 4. Parallel Trend Test (the TCZ on log monthly income).** Note: The baseline year is 1997, and the policy implementation year is 1998.

Second, our conceptual framework suggests that the TCZ policy may improve SWB by reducing air pollution and thereby enhancing health. Since our baseline results show a significant reduction in pollution, we next test the robustness of this health transmission channel. We re-estimate the effect of the TCZ policy on health (Table 5) and the effect of health on SWB (Table 6) under alternative specifications. Specifically, we examine the first stage of the mechanism (Fig 2) by estimating the following ordered logit model to determine whether the health status of affected residents improved after the implementation of the TCZ policy:

$$Health_{ihjpt} = \alpha_5 + \beta_5 \left( TCZ_j \times Post_t \right) + \eta_5 X_{ihjpt} + \phi_5 M_{hjpt} + \varphi_5 W_{jpt} + \omega \left( P \times f(t) \right) + \delta_j + \theta_t + \mu_{pt} + \varepsilon_{ihcjpt} \qquad (8)$$

Here, $Health_{ihjpt}$ is the self-reported health status of individual $i$, $X_{ihjpt}$ is a series of individual level control variables including years of education, age, marital status, gender, nationality, and urban/rural residence. Other variables are just as previously defined. Table 5 reports the marginal effects of the TCZ policy on the probabilities of being unhealthy, fair, and healthy, estimated from ordered logit model.

As shown in Column (1) of the table, after controlling appropriate factors and fixed effects, we find that the TCZ policy increases the probability of reporting good health by 3.3 percentage points. This outcome confirms that the TCZ policy does improve the health status of the affected residents, which is consistent with the research findings of Wang et al [73]. Specifically, their study found that the TCZ policy led to a 39% reduction in the four-week prevalence of diseases associated with air pollution.

**Table 5. The marginal effect of the TCZ Policy on health.**

|  | (1) | (2) | (3) |
|---|---|---|---|
| The Probability To Be Unhealthy | −0.004* | −0.004* | −0.003* |
|  | (0.002) | (0.002) | (0.002) |
| The Probability To Be Fair | −0.030* | −0.029* | −0.028* |
|  | (0.016) | (0.016) | (0.016) |
| The Probability To Be Healthy | 0.033* | 0.033* | 0.031* |
|  | (0.018) | (0.018) | (0.018) |
| P×Year dummy | YES | NO | NO |
| P×Post$_{1998}$ | NO | YES | NO |
| P×t | NO | NO | YES |
| P×t×t | NO | NO | YES |
| P×t×t×t | NO | NO | YES |
| Individual Control | YES | YES | YES |
| Family Control | YES | YES | YES |
| Weather Control | YES | YES | YES |
| Year FE | YES | YES | YES |
| City FE | YES | YES | YES |
| Province-by-year FE | YES | YES | YES |
| Observations | 10,603 | 10,603 | 10,603 |

Note: Table 5 presents the estimation results of Equation (8) eporting the marginal effects from the ordered logit model that measure the impact of the TCZ policy on individual health status. Columns (1)-(3) correspond to alternative specifications of the time function $f(t)$. Column (1) uses time fixed effects; Column (2) employs a post-policy dummy; and Column (3) adopts a third-order time polynomial. Robust standard errors are reported in parentheses (*** $p<0.01$, ** $p<0.05$, * $p<0.1$).

We then turn to the right half of Fig 2 and regress:

$$SWB_{ihjpt} = \alpha_6 + \beta_6 Health2_{ihjpt} + \delta_6 Health3_{ihjpt} + \gamma_6 lnWage_{ihjpt} + \eta_6 X_{ihjpt} + \phi_6 M_{hjpt} + \varphi_6 W_{jpt} + \delta_j + \theta_t + \mu_{pt} + \varepsilon_{ihjpt} \tag{9}$$

Here, $Health2_{ihjpt}$ is a binary variable that equals 1 if the respondent reports a health status of "Fair", and 0 otherwise. Similarly, $Health3_{ihjpt}$ equals 1 if the reported health status is "Good", and 0 otherwise. Table 6 presents the marginal effects of reporting fair health (relative to poor health) and good health (relative to poor or fair health) on the probabilities of being very unhappy, unhappy, fair, happy, and very happy, based on an ordered logit model that includes individual, household, and weather-related covariates, as well as fixed effects.

As shown in Column (2) of the table, relative to residents who report poor health, those reporting fair health are 17.6 percentage points more likely to report being "Very Happy". This outcome is significant at the 1% level. As shown in Column (3) of the table, compared to those who report fair health, individuals in good health are 10.7 percentage points more likely to report being "Very Happy".

Third, a potential concern is the mismatch in data-collection periods: the CHIP surveys were conducted after the TCZ policy shock, so WTP estimates based on CHIP may not equal WTP in the policy year. To address this concern, we re-estimate WTP using year-specific estimates. More specifically, according to equation (7), the WTP is determined by its two components: the ratio $\frac{\hat{\beta}_4}{\hat{\gamma}_4}$ and the average income $\overline{R}_{CHIP}$. For the first component, we first regress equations (3) and (4) by year, to see how $\frac{\hat{\beta}_4}{\hat{\gamma}_4}$ changes over time. Table 7 reports the by year results of equation (4).

**Table 6. The marginal effect of health on SWB.**

| | (1) | (2) | (3) |
|---|---|---|---|
| | **Fair vs. Poor** | **Good vs. Poor** | **Good vs. Fair** |
| The Probability To Be Very Unhappy | −0.003*** | −0.009*** | −0.006*** |
| | (0.001) | (0.001) | (0.000) |
| The Probability To Be Unhappy | −0.023*** | −0.058*** | −0.036*** |
| | (0.005) | (0.005) | (0.002) |
| The Probability To Be Fair | −0.041*** | −0.105*** | −0.064*** |
| | (0.008) | (0.008) | (0.003) |
| The Probability To Be Happy | −0.001* | −0.004* | −0.002* |
| | (0.001) | (0.002) | (0.001) |
| The Probability To Be Very Happy | 0.068*** | 0.176*** | 0.107*** |
| | (0.014) | (0.013) | (0.004) |
| Individual Control | YES | YES | YES |
| Family Control | YES | YES | YES |
| Weather Control | YES | YES | YES |
| Year FE | YES | YES | YES |
| City FE | YES | YES | YES |
| Province-by-year FE | YES | YES | YES |
| Observations | 23,187 | 23,187 | 23,187 |

Note: Table 6 presents the estimation results of Equation (9), reporting the marginal effects from the ordered logit model, which measure the impact of individual health status on SWB. Column (1) shows the marginal effect of health2 in Equation (9), while Column (2) shows the marginal effect of health3. Column (3) reports the difference between (1) and (2). All control variables marked YES in the table are included in the regression but their coefficients are omitted for brevity. Robust standard errors are reported in parentheses (*** p<0.01, ** p<0.05, * p<0.1).

**Table 7. The relationship between income, air quality and SWB in different years.**

| | (1) | (2) | (3) | (4) |
|---|---|---|---|---|
| Year | **2002** | **2007** | **2008** | **2013** |
| $SO_2$ | −0.018*** | −0.004 | −0.003 | −0.007** |
| | (0.005) | (0.004) | (0.004) | (0.003) |
| Ln(R) | 0.213*** | 0.079 | 0.140 | 0.094* |
| | (0.082) | (0.085) | (0.099) | (0.057) |
| $|\hat{\beta}_4/\hat{\gamma}_4|$ | 0.08 | 0.05 | 0.02 | 0.07 |
| $\overline{R}_{CHIP}$ | 910.69 | 1981.33 | 2661.25 | 2232.16 |
| WTP | 76.96 | 100.32 | 57.03 | 166.22 |

Note: Table 7 reports the year-by-year results of Equation (4), with Columns (1)-(4) corresponding to different years (total observations=23,187). For brevity, we only present the coefficients of $SO_2$, Ln(R). The sample mean of income ($\overline{R}_{CHIP}$) is directly calculated from the corresponding CHIP subsample, while the implied WTP is computed using the estimated ratio $|\frac{\hat{\beta}_4}{\hat{\gamma}_4}|$ and $\overline{R}_{CHIP}$. All variables included in Equation (4) are included in the regressions. Robust standard errors are reported in parentheses (*** p<0.01, ** p<0.05, * p<0.1).

As shown in the table, the $\frac{\hat{\beta}_4}{\hat{\gamma}_4}$ ratio. is broadly stable. However, the estimated coefficients for the years 2007 and 2008 are not statistically significant. Therefore, our subsequent analysis primarily focuses on the results from 2002 and 2013. To obtain a policy-year approximation we consider two scenarios. (i) The "Average scenario": we take the simple average of the 2002 and 2013 ratios (see Table 8). (ii) The "Linear interpolation scenario": we linearly interpolate between the 2002 and 2013 ratios to estimate the ratio for the policy year (this yields a ratio of 0.9; see Table 8). For the income component,

**Table 8. Results based on various assumptions.**

|  | Average Scenario | Linear Interpolation Scenario |
|---|---|---|
| $|\hat{\beta}_4/\hat{\gamma}_4|$ | 0.08 | 0.09 |
| $\overline{R}_{CHNS}$ | 743.9 | 743.9 |
| WTP | 59.1 | 65.6 |
| $Value(\triangle SWB)$ | 14.1 | 19.8 |

Note: Table 8 reports the marginal rate of substitution ($|\hat{\beta}_4/\hat{\gamma}_4|$) between air pollution and income under different assumptions, as well as the corresponding WTP calculated using the average income from the CHNS.

we use the average income of the subsample from affected areas in the CHNS data. The corresponding results are presented in Table 8.

As shown in Table 8, our conclusion is robust across scenarios: although the TCZ policy reduces local income, the monetized welfare gains from reduced pollution exceed the income losses, yielding a positive net effect on SWB.

Finally, we address the difference in geographic coverage between the two datasets. As noted in the data section, the CHIP surveys cover most provinces in mainland China, whereas the CHNS surveys include only nine provinces. This discrepancy raises the concern that the WTP estimates derived from CHIP may not accurately represent the preferences of residents in the CHNS sample. To mitigate this issue, we restrict the CHIP sample to the provinces that overlap with the CHNS.

We replicate the procedures used for Tables 7 and 8 and present the corresponding results in Tables 9 and 10. As shown in Table 10, the welfare gains from pollution reduction continue to exceed the associated income losses, implying that the TCZ policy generates a positive net monetary benefit for affected residents.

## Conclusion

The trade-off between economic development and environmental protection has long posed a major challenge worldwide, particularly for policymakers in developing countries. On the one hand, the environmental explanation of the Easterlin paradox highlights a key limitation of relying solely on income growth to enhance subjective well-being. It seems that to achieve the final goal of improving happiness, changes must be made to the widely adopted development path of

**Table 9. The relationship between income, $SO_2$ and SWB (restricted geographic area).**

|  | (1) | (2) | (3) | (4) |
|---|---|---|---|---|
| Year | 2002 | 2007 | 2008 | 2013 |
| $SO_2$ | −0.020*** | −0.004 | −0.013* | −0.010** |
|  | (0.006) | (0.005) | (0.007) | (0.004) |
| Ln(R) | 0.179* | 0.259** | 0.211 | 0.143* |
|  | (0.095) | (0.130) | (0.134) | (0.083) |
| $|\hat{\beta}_4/\hat{\gamma}_4|$ | 0.11 | 0.02 | 0.06 | 0.07 |
| $\overline{R}_{CHIP}$ | 773.41 | 1789.73 | 2278.88 | 2214.23 |
| WTP | 86.41 | 27.64 | 140.40 | 154.84 |

Note: Table 9 reports the year-by-year estimates of Equation (4) using the CHIP dataset restricted to provinces also covered by the CHNS (total observations = 9,413). Columns (1)-(4) present results for 2002, 2007, 2008, and 2013, respectively. As in Table 8, we report only the coefficients of $SO_2$, Ln(R). The sample mean of income ($\overline{R}_{CHIP}$) is directly calculated from the corresponding CHIP subsample, while the implied WTP is computed using the estimated ratio $|\hat{\beta}_4/\hat{\gamma}_4|$ and $\overline{R}_{CHIP}$. All variables included in Equation (4) are included in the regressions. Robust standard errors are reported in parentheses (*** p<0.01, ** p<0.05, * p<0.1).

**Table 10. Results based on various assumptions (restricted geographic area).**

|  | Average Scenario | Linear interpolation Scenario |
|---|---|---|
| $|\hat{\beta}_4/\hat{\gamma}_4|$ | 0.09 | 0.13 |
| $\overline{R}_{CHNS}$ | 743.9 | 743.9 |
| WTP | 67.6 | 94.4 |
| $Value(\triangle SWB)$ | 21.6 | 45.5 |

Note: Table 10 reports the marginal rate of substitution ($|\hat{\beta}_4/\hat{\gamma}_4|$) between air pollution and income under different assumptions for the restricted sample, along with the implied WTP based on the CHNS average income.

"pollution before treatment" in many developing regions. On the other hand, concerns remain prevalent in these regions that enforcing environmental regulations might slow economic growth and, consequently, reduce people's happiness, given that their economies are more vulnerable than those of developed countries. In this sense, empirical evidence on the environmental explanation of the Easterlin paradox, especially the ones targeted at developing countries and areas, have particular importance and strong policy implications.

In this research, we examine the environmental explanation of the Easterlin paradox in China, the world's largest developing economy, by exploring the effects of a large-scale environmental regulation program-the TCZ Policy- on SWB of affected residents using observations from two multi-round household surveys (one of which is a longitudinal survey that provides a good panel). By comparing changes in pollution and income between TCZ cities and non-TCZ cities before and after the policy implementation under a series of DID approaches, our research offers robust evidence on how income, environmental quality, and pollution control interact to shape SWB—an area that has been underexplored in previous studies due to the scarcity of longitudinal SWB data.

We find that the regulation has successfully mitigated air pollution in the implemented area, though at a cost of local income. Overall, the environmental benefits outweighed the income losses, yielding a positive net monetary effect for affected residents. Ceteris paribus, an average resident would be willing to forgo approximately ￥109.56 (2009 CNY) in monthly income for a 10,000-ton reduction in local $SO_2$ emissions. Therefore, the net SWB improvement that TCZ policy brought to this average resident is equivalent to an increase in their income by about ￥59.04 in terms of 2009 CNY. We view this as evidence supporting the Easterlin paradox in China. The validity of our method is verified by a series of diagnostic tests, including parallel trend tests and covariate balancing checks. The results remain robust when replacing pollution with health status as the treatment variable and when accounting for potential temporal confounders.

Our results suggest that even in developing countries, a well-implemented air pollution regulation can greatly enhance the subjective wellbeing of affected residents, especially those living in industrialized and wealthier areas.

## Supporting information

**S1 Appendix. Additional tables and figures, including regression results without the $P \times f(t)$ interaction term, parallel trend tests, and balancing tests for time-varying covariates.**
(DOCX)

## Author contributions

**Data curation:** Lianhua Li, Juan Huang.

**Formal analysis:** Yangjie Wang, Lianhua Li, Hongjie Qiang.

**Funding acquisition:** Yangjie Wang, Hongjie Qiang.

**Investigation:** Juan Huang.

**Methodology:** Lianhua Li, Juan Huang, Hongjie Qiang.

**Supervision:** Yangjie Wang.

**Visualization:** Juan Huang.

**Writing – original draft:** Juan Huang, Hongjie Qiang.

**Writing – review & editing:** Lianhua Li, Hongjie Qiang.

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
