## [Decision Letter · Decision Letter 0]

17 Sep 2024

Dear Dr. Qiang,

We look forward to receiving your revised manuscript.

Kind regards,

Shihe Fu, Ph.D.

Academic Editor

PLOS ONE

Journal requirements: 1. When submitting your revision, we need you to address these additional requirements. Please ensure that your manuscript meets PLOS ONE's style requirements, including those for file naming. The PLOS ONE style templates can be found at https://journals.plos.org/plosone/s/file?id=wjVg/PLOSOne_formatting_sample_main_body.pdf and https://journals.plos.org/plosone/s/file?id=ba62/PLOSOne_formatting_sample_title_authors_affiliations.pdf. 2. Note from Emily Chenette, Editor in Chief of PLOS ONE, and Iain Hrynaszkiewicz, Director of Open Research Solutions at PLOS: Did you know that depositing data in a repository is associated with up to a 25% citation advantage (https://doi.org/10.1371/journal.pone.0230416)? If you’ve not already done so, consider depositing your raw data in a repository to ensure your work is read, appreciated and cited by the largest possible audience. You’ll also earn an Accessible Data icon on your published paper if you deposit your data in any participating repository (https://plos.org/open-science/open-data/#accessible-data). 3. We note that the grant information you provided in the ‘Funding Information’ and ‘Financial Disclosure’ sections do not match.  When you resubmit, please ensure that you provide the correct grant numbers for the awards you received for your study in the ‘Funding Information’ section. 4. We note that you have indicated that there are restrictions to data sharing for this study. PLOS only allows data to be available upon request if there are legal or ethical restrictions on sharing data publicly. For more information on unacceptable data access restrictions, please see http://journals.plos.org/plosone/s/data-availability#loc-unacceptable-data-access-restrictions.  Before we proceed with your manuscript, please address the following prompts: a) If there are ethical or legal restrictions on sharing a de-identified data set, please explain them in detail (e.g., data contain potentially identifying or sensitive patient information, data are owned by a third-party organization, etc.) and who has imposed them (e.g., a Research Ethics Committee or Institutional Review Board, etc.). Please also provide contact information for a data access committee, ethics committee, or other institutional body to which data requests may be sent. b) If there are no restrictions, please upload the minimal anonymized data set necessary to replicate your study findings to a stable, public repository and provide us with the relevant URLs, DOIs, or accession numbers. For a list of recommended repositories, please seehttps://journals.plos.org/plosone/s/recommended-repositories. You also have the option of uploading the data as Supporting Information files, but we would recommend depositing data directly to a data repository if possible. We will update your Data Availability statement on your behalf to reflect the information you provide.

Reviewers' comments:

Reviewer's Responses to Questions

**Comments to the Author**

1. Is the manuscript technically sound, and do the data support the conclusions?

Reviewer #1: Partly

Reviewer #2: Partly

2. Has the statistical analysis been performed appropriately and rigorously?

Reviewer #1: Yes

Reviewer #2: No

3. Have the authors made all data underlying the findings in their manuscript fully available?

Reviewer #1: Yes

Reviewer #2: No

4. Is the manuscript presented in an intelligible fashion and written in standard English?

Reviewer #1: Yes

Reviewer #2: No

Reviewer #1: This paper proposes the environmental explanation to the Easterlin paradox. Using the survey data of CHNS and CHIP, the paper estimates the effect of Tow Control Zone policy on households’ subjective wellbeing with a difference-in-difference approach, and further calculate the households’ willingness to pay for the environment. In general, I feel that the paper is well-written, the ideas is clear and easy to follow. Yet, there are some concerns regarding the research design and the empirical approach that need to be addressed to pass on to the next step. Following are some of my comments.

1.Prediction of household subjective wellbeing (SWB). To derive the dependent variable of SWB, which is absent in the CHNS data, the authors construct a SWB function (model 1) to predict the SWB probability using another CHIP data. A key assumption underlying the approach is that the estimated SWB function is national representative, so that it can be transferred from one data to another. However, given the fact that the two datasets have different geographical coverage and time periods, it is hard for me to believe that households in the two datasets have the same preferences, and that you can transfer the SWB function from one group of households to another group of households. You should further justify the representativeness of the CHIP data and the appropriateness of your approach.

2.Construction of the SWB function. In the model 1, it is clear that the SWB is predicted from the household health status, income, and a number of socio-economic characteristics. Since you also controlled for the same set of socio-economic characteristics in model (2), the predicted SWB equals to a composite indicator of health status and income. But in the follow-up analysis, you explain the effect of TCZ policy on SWB from the perspective of air pollution and income as in model (3) and (4). I wonder why don’t you just predict the SWB from the air pollution and income in model (1) to make it more consistent with the follow-up analysis?

3.Construction of the fake data. I feel it odd to inflate the data by expanding 1 observation to 100, should this trick artificially creates some problems of autocorrelation? Why don’t you just use a simple linear probit model in the first step to predict the SWB for households with the original CHNS data? This is more clear and simple for implementation.

4. Identification of the TCZ effect. The key assumption of difference-in-difference approach is the parallel trend assumption. However, it is un clear how do you test this assumption from the main text, and the results in table 4 are doubtful without any figures to show the parallel trend. Moreover, it is also important to test the parallel trend assumption for the follow-up analysis of air pollution and income. These test results could be put into appendix.

5. Misunderstanding in the heterogeneity analysis. How do you implement the heterogeneity analysis in section 5.2? Should you divide the full sample into subsamples or use interaction terms? It is unclear from reading the text, and in Table 5-7, it seems that you just simply add the classification variables in the model. You should clarify your approach.

6.The endogeneity problem. You mention that two instruments are employed to address the endogeneity issue of air pollution in Equation (8). I guess that you mean the Equation (5). If you think that the air pollution here is endogenous, it is also the concern for income. You should address the endogeneity problems of both air pollution and income with additional IVs. You should also carefully discuss the relevance and exclusive restriction of your IVs and present the first stage results of the IV estimations, which is now absent in the text.

7. Contribution of the paper. In the literature, there are a number of papers to estimate the impact of TCZ policy in China. Why your paper is novel compared to other TCZ studies? You should strengthen your contribution to the literature in the introduction.

Reviewer #2: The authors shed light on the Easterlin Paradox, a phenomenon in which economic growth does not improve the SWB of people. One of the typical explanations for this paradox is that environmental quality often deteriorates as economy grows, and the potential improvement in the SWB from economic growth is off-set by the SWB loss owing to deteriorated environmental quality. The authors examined the inverse story of this explanation, that is, whether and how an environmental regulation, which can reduce environmental pollution but hinder economic activities, affects SWB of people, using data from China and utilizing the TCZ policy implemented in 1998. The authors further estimated the WTP for pollution reduction and the monetary value of the TCZ policy. The results showed that the TCZ policy improved the SWB of people, reduced SO2 pollution, and lowered wage incomes. The authors concluded that the abovementioned inverse story of the Easterlin Paradox was supported.

While the question addressed is interesting, I have concerns on the scientific soundness and appropriateness of the analyses. The main results and conclusion of the study is entirely dependent on the “fake” dataset. My main concerns are 1) the estimation results to generate the fake dataset is only partially disclosed; and 2) the sample size is expanded by 100 times during the process, and this is likely to artificially lower the standard errors of the main results (detailed comments below).

Let me summarize their process to generate the fake dataset. The authors first estimated the equation (equation 1) to explain the SWB level (happy, fair, unhappy) by ordered logit based on the CHIPS data, which covers only the post-TCZ periods (called first stage). Then they predicted the probabilities that each CHNS sample individuals, which cover pre- and post-TCZ periods, has the three levels of SWB, extrapolating the estimated equation (1) to the CHNS data. Finally, the authors employed 1:100 expansion, in which each CHNS individual appears 100 times, to convert the predicted probabilities into categorical choice variable (if the probability of SWB=3 is 0.3, SWB=2 is 0.4, and SWB=1 is 0.3, then this individual appears 100 times in data, 30 times with SWB=3, 40 times with SWB=2, and 30 times with SWB=1). I understand the problem that the SWB data covering both the pre- and post-TCZ policy are unavailable. However, there are several major concerns with this procedure as follows.

1. The biggest concern is that the authors did not report the details of the estimation results to generate fake dataset, although the authors provided a clear methodological process. Authors graphically demonstrated the marginal effects of income and self-reported health status on the probabilities to choose three levels of SWB in the CHIPS data but omitted other details, noting “instead of reporting the meaningless estimates on coefficients (p31).” The problem is not either the coefficients or marginal effects are better. It is that all other information is not disclosed. More information is needed, such as the significances and signs of other variables (or ME), goodness of fit and/or overall explanation power, sample size and sample-selection criteria, etc. Although the overall results of this study were interesting, it was hard to agree or disagree with them if the derivation of the fake dataset is mostly kept in a blackbox.

Further concerns are on their 1:100 expansion method. The authors noted “that this trick is just to deal with the problems caused by the particular characteristic of ordered choice model, otherwise, what we do is nothing different from predicting a missing SWB and applying it for further research, which is common in researches facing incomplete data problems (see Little and Ann 2004, Kang and Schafer 2007, Penn 2009).” However, it is not that simple.

2. It does not seem that the authors dealt with the decreased standard errors owing to the increased sample size by 100-time appearance of each individual. That procedure would lower the standard errors and the t-values of the variables would be inflated. The authors did not state anything about this issue, and did not cite any study that utilize similar 1:100 expansion. I suspect that, after appropriate treatment of the standard errors, some of the coefficients that have values close to zero will become insignificant.

3. Further, why did the authors use non-linear model in the first stage but linear model in the second-stage estimations, although the dependent variable is the same SWB? Clearly, the SWB is a categorical measure. But in the second stage, the authors anyway mainly used “traditional fixed effect DID” and treated the SWB as a continuous variable. 1) If the non-linear model is preferable, then the second stage should also use non-linear model. 2) But if a linear-model is sufficient, then the authors can use linear model in the first stage as well and do not need to apply 1:100 expansion. What if the authors use a linear regression in the first stage, predict SWB^hat of CHNS individuals (which will be mostly non-integer values), and regress SWB^hat in the second stage without 1:100 expansion?

4. The authors cited three papers, but they basically focus on cases where key variables are missing for subsamples. But in this study, the key variable, SWB, is completely missing in the CHNS data and predicted from CHIP data. I felt that “nothing different” is not reasonable. For example, an underlying assumption to extrapolate the CHIP-based equation to the CHNS data is that the preferences of people in the CHIPS (starting from 2003) and the CHNS (1991–2006) are unchanged. But is it justifiable? In particular, a marginal effect of a one-yuan increase in income on the SWB could be different over time.

5. The main results and various robustness check and heterogeneity analyses showed that the TCZ policy improved the SWB. Then the authors argued that the positive effect is “evidence supporting the environmental explanation of the Easterlin paradox from an inverse story logistic (p32).” But the positive coefficient itself does not explain anything about the Easterlin Paradox, because at this moment, the authors have not provided any evidence that the TCZ policy simultaneously improved environmental quality and lowered economic welfare. The authors argue so conceptually in Figure 3 (p21), but the summary statistics (p25) rather suggest the opposite: the TCZ policy improves both the environmental quality and wage rates. It is Table 8 (pp43¬-44) where the authors provided evidence that the TCZ policy simultaneously improved environmental quality and lowered economic welfare for the first time (although several questions exist for this result, see below). Therefore, under the current structure, it was hard to agree with the abovementioned claim in p32. In the conclusion section, the authors explain the results of Table 8 first and then the impact of the TCZ policy on the SWB. The results section should proceed in the same way.

Section 5.3 is directly related to the inverse story of the Easterlin Paradox. While the WTP calculation method itself is fine, the estimation is questionable.

6. In column (26), the sample size is 1245170, meaning that the fake dataset was used. But in this estimation, because the SWB is not used, the authors should directly use the original CHNS dataset and do not have to use the fake dataset in which the same individuals appear 100 times. I wonder if the TCZxPost remains significant if the original CHNS dataset is used. Indeed, the summary statistics showed that the wage rate grew faster in the TCZ cities than in the non-TCZ cities, suggesting that the TCZ policy increased wage rate and lowered the pollution level. Maybe the effect of TCZ policy was reverted in column (26) after controlling for other factors (then it supports the inverse story of Easterlin Paradox). But the significant coefficient may just reflect the small standard errors caused by the 1:100 expansion. In sum, because there is no need to use the fake dataset, the authors should try this estimation with the original CHNS dataset.

7. In column (27), the SWB is regressed to SO2 level and log(wage) based on the fake dataset. But because the TCZ policy variable is not used this time, the authors can use the original CHIP dataset for this analysis. Do these coefficients remain significant if the original CHIPS data are used and the sample size is not artificially increased by 100 times?

8. Further, the estimation of column (27) uses an IV. The IV itself is fine (used widely in the literature), I wonder why endogeneity matters. The dependent variable this time, SWB^hat, is basically predicted from the observable characteristics and unobservable factors cannot influence SWB^hat. Even if the authors continue using an IV, further information is needed, such as first-stage results, F-value, etc.

9. The authors used the 1:100 expansion, so the sample size should be multiples of 100. But I saw the sample size of 1245170, 1246489, etc. Why? Is the CHIPs data also used as samples without multiplication?

Other comments

10. The literature review and background sections were too lengthy, spanning from p7–p21. The authors should shorten these parts by reducing irrelevant information. Further, in these sections, citation is incorrect or simply missing in the reference list. For example, DiMaria and Sarracino (2019), Naghdi et al. (2014), and Stelzner (2021) are the in-text citation (pp7–8), but according to the reference list, they should be DiMaria and Sarracino (2020), Naghdi et al. (2021), and Stelzner (2022). In p15, World Bank (2015), NBS (1991; 2007), and Ministry of Ecology and Environment, PRC, 1996) are cited, but they do not appear in the reference list. These are just examples, and there are other papers inaccurately cited. Careful re-checking is needed.

11. In pp 19–21 where the theoretical channels are discussed, the authors explain the possibilities that environmental regulation can positively affect income, not only the possibility that the regulation hinders income. It is nice. But I think there is an additional channel: an improved health and environment quality increase the income by improving labor productivity and labor supply amount. There are a lot of studies examining this point, such as

Aragón, F.M., Miranda, J.J., Oliva, P., 2017. Journal of Environmental Economics and Management 86, 295–309.

Borgschulte, M., Molitor, D., Zou, E.Y., (forthcoming). The Review of Economics and Statistics.

Fu, S., Viard, V.B., Zhang, P., 2022.

Although this channel is not the main one this study considers, a brief mentioning to this channel may improve the clarity of the conceptual flow.

12. Estimation equations and variable notations need revision. Equation (1) is a linear equation and is not an ordered-choice equation. Equation (2) is the main second-stage equation, but it needs the term for fixed effects if fixed effects model is actually used in the results.

13. In p 27, the authors wrote “We use the ordered logit model to estimate Equation (1), and the result tells us, conditional on income, health and other individual, household and city characteristics,what’s the probability an individual will say good, fair, or poor when asked: “Do you feel happy now?”. To make the two data sets consistent, we convert the four-level rank in CHIP data to three levels: excellent is converted into good, good is converted into fair, and poor and very poor are converted into poor.” But what variable are you talking about? Context-wise, it sounds like the SWB level was converted, but this cannot be true because CHNS data do not have SWB and consistency does not matter? Are you talking about household income?

14. PSM is used in column (2), but more information is needed, such as the equation to estimate the PSM, matching method (nearest neighbor? Kernel? Or other?).

15. In Table 7, cities with high proportion of primary industry are labelled as “agricultural.” But it is counterintuitive that the TCZ policy worsened the SWB of people. But I think primary industry also includes mining. So it should be labelled as “agricultural and mining”. And it is very intuitive that the TCZ policy worsened the SWB of people in mining sector.

16. In p43, “In estimating Equation (8), we use […] we report the results in Column (27)”. But in equation (8), there is nothing to additionally estimate: beta^hat3 and beta^hat4 are estimated based on equations (3) and (4), and WTP^hat is calculated from equations (5)–(7). Perhaps the author examined an additional equation (not equation 8) and showed the results in column (27). Careful revision of the explanation is needed.

17. Further, the notation of ΔWTP^hat is a bit problematic. WTP^hat can take only the values of 1,2,3. So ΔWTP^hat being 649.499 confuses readers. ΔWTP^hat stands for “the monetized value of WTP changes”, so some other notation will be better.

18. Data availability statement states only the URL of the CHIP (which I could not access for some reason), but the authors used various other datasets. Further, these data sources are not cited in the reference (CHNS, NASA’s satellite data, etc). They should be cited. Data availability and reasons that data cannot be shared by the authors should be more clearly stated, instead of just writing the URL for CHIP dataset.

19. Language editing is needed. While overall the manuscript is written in good English, there are quite a lot of small mistakes (e.g. German instead of Germany). Wordy and lengthy parts can be shorten.

**Do you want your identity to be public for this peer review?** For information about this choice, including consent withdrawal, please see our Privacy Policy

Reviewer #1: **Yes:** Huanxiu Guo

Reviewer #2: No

---

## [Author Response · Author response to Decision Letter 1]

5 Mar 2025

Dear editor,

Thank you for your email of 18 September 2024 concerning our manuscript “Wealth, Health, and Happiness: An Inverse Story of the Easterlin Paradox in China” (PONE-D-23-32960). We would like to thank you for the full consideration of our paper and sending us the comments of two reviewers. We have revised the paper according to yours and the reviewers’ suggestions, and mark the changed text in red.

The major revisions are summarized as follows:

1. We made a series of modifications to the methodology used in our research, specifically:

1) We no longer predict SWB across datasets, instead, we adopted the three steps approach suggested by you to evaluate the effect of TCZ policy.

2) We explore more details about how WTP changes over time and use the knowledge to provide robustness checks regarding the concern about period difference across data sets.

3) We used average log wage for staffs who work in the same industry and the same province as instrument variables for income.

2.We made a series of modifications in the Literature review and Background components, including:

1) We placed the literature review before the background as suggested.

2) We deleted several less relevant contents from these two parts.

3) We added several important literatures as suggested.

3.We had a professional editing of the paper. Now the writing has been substantially improved.

With this letter, we have resubmitted our manuscript which we would like to be considered for publication in Plos one. In order to facilitate the review of our revisions, we are attaching a detailed, comment by comment response to the two reviewers’ concerns.

If you need anything else, please do not hesitate to contact us.

Kind regards,

All authors

Response of Authors to the Comments of Editor.

Editor’s Comments:

1. The topic is interesting and you have

done a lot of work, but using CHIPS data's SWB information to predict SWB in

CHNS is problematic. Actually, you do not have to do so. I have the following

proposal to answer your research question without imputing SWB:

estimate the effect of two-control zone policy on air pollution, wage, and health, this can be done using only CHNS;

estimate the trade off between pollution (or health) and income, using only CHIPS, following Levison (2012).

Combining both 1 and 2, you can still back up the net effect of (net willingness to pay for) the two-control zone policy, as you did in Table 8.

You can use PSM method to predict individual's SWB in CHNS without inflating the sample size, and check if the effect for TCZ policy on imputed SWB matches your calculation in step 3, if you wish. But I don't think this is a must.

Authors’ Response:

Thank you for your valuable suggestions. That provides us with a good framework to address reviewers’ key concerns.

As suggested, we now choose to use a three-stage approach to combine information from CHNS data and CHIP data and to evaluate the total effect of TCZ policy on SWB.

“First, we use a typical difference in difference (DID) approach to estimate the effect of the TCZ policy on air pollution and residential income of affected cities.

For air pollution, we regress:

P_jpt=α_1+β_1 (〖TCZ〗_j×〖Post〗_t )+φ_1 W_jpt+δ_j+θ_t+μ_pt+ε_jt (1)

Here, P_jpt is the air pollution of city j (which locates in province p) at time t, measured by average ambient concentration of SO2 at the city level. 〖TCZ〗_j is a dummy variable indicating whether the observation is from a city covered by the TCZ policy (takes the value of 1 if it is) while 〖Post〗_t is a dummy variable indicating whether the observation is from the post-policy period (takes the value of 1 in the year 1998 and so after). β _1 is then the parameter of interest which illustrates the effect of TCZ policy on city air pollution. In this regression, we control a series of city level meteorological factors and economic factors such as temperature, precipitation and logarithm of local GDP, the city fixed effect δ_j, the time fixed effect θ_t, and the combined fixed effect at a higher dimension, i.e., μ_pt=〖Province〗_j×〖time〗_t. ε_jpt is a heteroskedasticity-robust standard error term.

For residential income, we regress:

R_ihcjpt=α_2+β_2 (〖TCZ〗_j×〖Post〗_t )+η_2 X_ihcjpt+ϕ_2 M_chjpt+ν_2 D_cjpt+φ_2 W_jpt+δ_j+θ_t+μ_pt+ε_ihcjpt (2)

Here, R_ihcjpt is the average monthly income of individual i at time t, where the individual is from household h and lives in the community c of city j in province p. X_ihcjpt is a series of individual level control variables including the education year, gender, ethnicity and age. M_chjpt is a series of household level control variables including family size and the household’s income. D_cjpt is a series of community level control variables including the population density and the market composition index. W_jpt, δ_j, θ_t and μ_pt are the same as in equation (1), and β _2 is the parameter of interest which indicates the effect of TCZ policy on monthly income of affected individuals. The above two regressions are estimated using the CHNS data.

Second, following the wisdom of Levinson (2012), Ambrey et al. (2014) and Zhang et al. (2017a, 2017b), we estimate residents’ WTP for improved air quality by exploring their trade-offs between economic benefit and environmental quality subject to the constraint that their SWB stay the same. Since both air quality and income might be correlated with factors which also influence SWB, we apply a series of instrumental variables to deal with potential endogeneity issues. We also use the methods adopted by the ordered choice models to take care of the potential problems from the discrete choice issues following Dolan et al. (2008), Diener et al. (2018) and Clark (2018).”

“We use a 2SLS approach to estimate this IV model. The 1st stage regressions are:

P_jpt=α_3+β_3 〖Days〗_jpt+γ_3 〖Wind〗_jpt+φ_3 W_jpt+δ_j+θ_t+μ_pt+ε_jt (3)

and:

lnR_ihjkpt=α_4+β_4 ln〖Wage〗_kpt+η_4 X_ihjt+ϕ_4 M_hjpt+φ_4 W_jpt+δ_j+θ_t+μ_pt+ε_ihjpt (4)

In equation (3), 〖Days〗_jpt is the weighted average of number of days in the year when the second layer of the atmosphere is warmer than the first layer and that when the third layer of the atmosphere is warmer than the first layer. The weight is 1:1. 〖Wind〗_jpt is the annual average wind speed of the city. In equation (4), lnR_ihjkpt is the log income of individual i at time t, where individual i is from household h, lives in the community c of city j in province p, and works for industry k. ln〖Wage〗_kpt is the average log wage for staffs who work in the same industry and the same province as individual i. Other control variables are the same as in equation (1) and (2).

In the 2nd stage, we regress the following ordered choice model:

〖SWB〗_ihjkpt=α_5+β_5 P _jpt+γ_5 ln⁡〖R _ihjkpt 〗+η_5 X_ihjpt+ϕ_5 M_hjpt+φ_5 W_jpt+δ_j+θ_t+μ_pt+ε_ihjpt (5)

Here, P _jpt are predicted through equation (3) while ln⁡〖R _ihjkpt 〗 are predicted from equation (4).”

“according to Levinson (2012), Ambrey et al. (2014) and Zhang et al. (2017a, 2017b), the WTP of affected residents on improved air quality (measured by reduced SO2 concentration) can be calculated by:

(WTP) =∂R/∂P |■(@@dSWB=0)┤=-R β _5/γ _5

Here, R is the average income of the sample. Equation (3)~(7) are estimated using the CHIP data.

Finally, with influence of TCZ policy on air pollution (β _1) and monthly income of affected residents (β _2) estimated in the first step and the residents’ WTP for less air pollution estimated in the second step, we can calculate the average net money value of TCZ policy on affected residents through:

Value(∆SWB)=β _1 (WTP) +β _2”

The related content can be found in page 22-27 of our replenished manuscript.

Response of Authors to the Comments of Reviewer #1

Reviewer’s Comments:

GENERAL COMMENTS

This paper proposes the environmental explanation to the Easterlin paradox. Using the survey data of CHNS and CHIP, the paper estimates the effect of Tow Control Zone policy on households’ subjective wellbeing with a difference-in-difference approach, and further calculate the households’ willingness to pay for the environment. In general, I feel that the paper is well-written, the ideas is clear and easy to follow. Yet, there are some concerns regarding the research design and the empirical approach that need to be addressed to pass on to the next step. Following are some of my comments.

Authors’ Response:

Thank you for all your nice comments and suggestions. Through the revisions, this paper has been substantially improved. We have revised the paper according to your comments. We hope you would find the revisions satisfactory.

MAJOR COMMENTS

1) Prediction of household subjective wellbeing (SWB). To derive the dependent variable of SWB, which is absent in the CHNS data, the authors construct a SWB function (model 1) to predict the SWB probability using another CHIP data. A key assumption underlying the approach is that the estimated SWB function is national representative, so that it can be transferred from one data to another. However, given the fact that the two datasets have different geographical coverage and time periods, it is hard for me to believe that households in the two datasets have the same preferences, and that you can transfer the SWB function from one group of households to another group of households. You should further justify the representativeness of the CHIP data and the appropriateness of your approach.

Authors’ Response:

Thank you for highlighting the potential issues regarding the use of CHIP data to construct the SWB function and its application to CHNS data. We appreciate your valuable suggestions, and we agree that trying to combine information from two data sets with different geographic areas and periods is challenging.

After carefully discussions around opinions from the editor and two reviewers, we have made following changes to our empirical strategy in identifying the comprehensive effects of TCZ policy on SWB.

First, we no longer trying to predict the SWB for CHNS interviewers using CHIP data. Instead, following the suggestion from the editor, we use a less complex approach to draw information from CHIP data and CHNS data. The main process are as follows:

“First, we use a typical difference in difference (DID) approach to estimate the effect of the TCZ policy on air pollution and residential income of affected cities.

For air pollution, we regress:

P_jpt=α_1+β_1 (〖TCZ〗_j×〖Post〗_t )+φ_1 W_jpt+δ_j+θ_t+μ_pt+ε_jt (1)

Here, P_jpt is the air pollution of city j (which locates in province p) at time t, measured by average ambient concentration of SO2 at the city level. 〖TCZ〗_j is a dummy variable indicating whether the observation is from a city covered by the TCZ policy (takes the value of 1 if it is) while 〖Post〗_t is a dummy variable indicating whether the observation is from the post-policy period (takes the value of 1 in the year 1998 and so after). β _1 is then the parameter of interest which illustrates the effect of TCZ policy on city air pollution. In this regression, we control a series of city level meteorological factors and economic factors such as temperature, precipitation and logarithm of local GDP, the city fixed effect δ_j, the time fixed effect θ_t, and the combined fixed effect at a higher dimension, i.e., μ_pt=〖Province〗_j×〖time〗_t. ε_jpt is a heteroskedasticity-robust standard error term.

For residential income, we regress:

R_ihcjpt=α_2+β_2 (〖TCZ〗_j×〖Post〗_t )+η_2 X_ihcjpt+ϕ_2 M_chjpt+ν_2 D_cjpt+φ_2 W_jpt+δ_j+θ_t+μ_pt+ε_ihcjpt (2)

Here, R_ihcjpt is the average monthly income of individual i at time t, where the individual is from household h and lives in the community c of city j in province p. X_ihcjpt is a series of individual level control variables including the education year, gender, ethnicity and age. M_chjpt is a series of household level control variables including family size and the household’s income. D_cjpt is a series of community level control variables including the population density and the market composition index. W_jpt, δ_j, θ_t and μ_pt are the same as in equation (1), and β _2 is the parameter of interest which indicates the effect of TCZ policy on monthly income of affected individuals. The above two regressions are estimated using the CHNS data.

Second, following the wisdom of Levinson (2012), Ambrey et al. (2014) and Zhang et al. (2017a, 2017b), we estimate residents’ WTP for improved air quality by exploring their trade-offs between economic benefit and environmental quality subject to the constraint that their SWB stay the same. Since both air quality and income might be correlated with factors which also influence SWB, we apply a series of instrumental variables to deal with potential endogeneity issues. We also use the methods adopted by the ordered choice models to take care of the potential problems from the discrete choice issues following Dolan et al. (2008), Diener et al. (2018) and Clark (2018).”

“We use a 2SLS approach to estimate this IV model. The 1st stage regressions are:

P_jpt=α_3+β_3 〖Days〗_jpt+γ_3 〖Wind〗_jpt+φ_3 W_jpt+δ_j+θ_t+μ_pt+ε_jt (3)

and:

lnR_ihjkpt=α_4+β_4 ln〖Wage〗_kpt+η_4 X_ihjt+ϕ_4 M_hjpt+φ_4 W_jpt+δ_j+θ_t+μ_pt+ε_ihjpt (4)

In equation (3), 〖Days〗_jpt is the weighted average of number of days in the year when the second layer of the atmosphere is warmer than the first layer and that when the third layer of the atmosphere is warmer than the first layer. The weight is 1:1. 〖Wind〗_jpt is the annual average wind speed of the city. In equation (4), lnR_ihjkpt is the log income of individual i at time t, where individual i is from household h, lives in the community c of city j in province p, and works for industry k. ln〖Wage〗_kpt is the average log wage for staffs who work in the same industry and the same province as individual i. Other control variables are the same as in equation (1) and (2).

In the 2nd stage, we regress the following ordered choice model:

〖SWB〗_ihjkpt=α_5+β_5 P _jpt+γ_5 ln⁡〖R _ihjkpt 〗+η_5 X_ihjpt+ϕ_5 M_hjpt+φ_5 W_jpt+δ_j+θ_t+μ_pt+ε_ihjpt (5)

Here, P _jpt are predicted through equation (3) while ln⁡〖R _ihjkpt 〗 are predicted from equation (4).”

“according to Levinson (2012), Ambrey et al. (2014) and Zhang et al. (2017a, 2017b), the WTP of affected residents on improved air quality (measured by reduced SO2 concentration) can be calculated by:

(WTP) =∂R/∂P |■(@@dSWB=0)┤=-R β _5/γ _5

Here, R is the average income of the sample. Equation (3)~(7) are estimated using the CHIP data.

Finally, with influence of TCZ policy on air pollution (β _1) and monthly income of affected residents (β _2) estimated in the first step and the residents’ WTP for less air pollution estimated in the second step, we can calculate the average net money value of TCZ policy on affected residents through:

Value(∆SWB)=β _1 (WTP) +β _2”

The related content can be found in page 22 - 27 of our replenished manuscript.

Second, we adopted a series of robustness checks regarding the inconsistency in time and area between the two data sets.

For time inconsistency:

“Fourth, one concern is that the CHIP data are collected in a period different from the year of the policy shock, therefore the WTP estimated from CHIP data may be different from that of the policy shock year, if WTP changes over time. In this sense, we recalculate WTP by taking account of this concern. More specifically, according to equation (7), the WTP is determined by its two components: the ratio β _5/γ _5 and the average income R . For the first component, we first regress equation (3) ~ (5) by year, to see how β _5/γ _5 changes overtime. Table 9 reports the by year results of equation (5).”

“As shown in the table, the β _5/γ _5 ratio takes a special high value in the year 2002(1.04), then it dramatically decreased to a level around 0.20 in the year 2007, 2008, and 2013, and does not show a clear time trend then. There are two possibilities which might lead to such phenomena. First, it is possible that before 2002, the ratio is fluctuating around 1.04, and then a structural change in the correlation happens sometime between 2002 and 2007 which result in a new stable ratio around 0.2. In this case, we should use 1.04 to approximate the ratio in th

---

## [Decision Letter · Decision Letter 1]

4 Apr 2025

Dear Dr. Qiang,

Thank you for submitting your manuscript to PLOS ONE. After careful consideration, we feel that it has merit but does not fully meet PLOS ONE’s publication criteria as it currently stands. Therefore, we invite you to submit a revised version of the manuscript that addresses the points raised during the review process.

We look forward to receiving your revised manuscript.

Kind regards,

Chih-Wei Tseng

Academic Editor

PLOS ONE

Additional Editor Comments:

Kindly address each of the reviewer’s comments individually, as outlined in the attached document.

Reviewers' comments:

Reviewer's Responses to Questions

**Comments to the Author**

Reviewer #1: All comments have been addressed

Reviewer #2: All comments have been addressed

2. Is the manuscript technically sound, and do the data support the conclusions?

Reviewer #1: Yes

Reviewer #2: Partly

3. Has the statistical analysis been performed appropriately and rigorously?

Reviewer #1: Yes

Reviewer #2: Yes

4. Have the authors made all data underlying the findings in their manuscript fully available?

Reviewer #1: Yes

Reviewer #2: No

5. Is the manuscript presented in an intelligible fashion and written in standard English?

Reviewer #1: Yes

Reviewer #2: Yes

Reviewer #1: (No Response)

Reviewer #2: (No Response)

**Do you want your identity to be public for this peer review?** For information about this choice, including consent withdrawal, please see our Privacy Policy

Reviewer #1: **Yes:** Huanxiu Guo

Reviewer #2: No

---

## [Author Response · Author response to Decision Letter 2]

19 May 2025

(We have uploaded a Word version of the response letter, in which the replies are presented more clearly)

Comments to the Author

GENERAL COMMENTS

The authors made a significant improvement to the manuscript. The methods are improved, and the results are presented in a logically organized manner compared to the previous manuscript.

Whereas I appreciate the improvement made by the authors, I still see several points that need to be clarified. Some of the points are related to methodologies, while some others are related to inaccurate, confusing explanations (and sometimes explanations are lacking, old sentences from the previous manuscript are remaining, etc). Thank you very much for your constructive comments and suggestions.

We have revised the manuscript carefully in response to your comments. We hope you find the revisions satisfactory.

MAJOR COMMENTS

1)Why don’t you use FE or RE for CHNS dataset?

The authors noted that CHNS data is a panel dataset. They further stated “the longitudinal essential of the data also allows us to better control for unobserved individual heterogeneity and the potential selection bias problem” in Section 4.1. However, based on the methods and results, the authors seemingly did not use individual/household FE to control these confounding factors. If there is a reason for not using the individual fixed effects and treating the CHNS panel as just a pooled dataset, then the authors should describe so and justify it. The authors also need to delete some of the sentences that sounds as if the authors are using FEs. Otherwise, it is straightforward to use individual FEs. Thank you very much for your insightful comment. We sincerely appreciate your suggestion regarding the use of individual fixed effects (FEs) for the CHNS dataset.

In our initial submission, we did not include individual fixed effects in the CHNS sample regressions, which may have led to potential concerns regarding unobserved heterogeneity. We acknowledge this as a methodological omission and have now addressed it in our revised manuscript. Specifically, we have re-estimated the relevant regressions using individual fixed effects, implemented via the “reghdfe” command in Stata. By including individual FEs, we are able to control for all time-invariant individual-specific characteristics, thereby improving the robustness of our estimates and better addressing potential selection bias.

In addition, as you rightly pointed out in your subsequent comment, we observed instability and theoretical inconsistency in the coefficient estimates of the GDP variable, which is likely due to multicollinearity. In response, we have removed GDP from all relevant equations to enhance the model’s clarity and focus.

Specifically, we add τ_i which represents individual fixed effect in equation (2). We also revised the related descriptions in Section 4.1:

“X_ihcjpt is a series of time-varying individual level control variables including years of education, age, and marital status. M_hcjpt is a series of household level control variables including family size and the household’s income. D_cjpt is a series of community level control variables including marketization score and transportation infrastructure score. τ_i is the individual fixed effect”.

Thank you again for your constructive feedback, which has helped us significantly improve the methodological rigor of our work.

2) Data description

Apart from the comment above, the description of the data sources is confusing. The authors wrote “We combine data from the following sources to generate a 6 years panel across 8 provinces, 40 cities (21 cities are TCZ cities and 19 are non-TCZ cities), 5,058 households and 19,538 individuals to accomplish our research targets” at the beginning of Section 4.1. But this is simply inaccurate. Firstly, the estimations based on the CHNS data have the sample sizes around 23,000 (Table 3), whereas those based on the CHIP data have the sample size of 20,887 (Table 4). Secondly, the area coverage and timing of these surveys are different. CHNS cover 10 provinces, whereas CHIP cover all provinces.

As the authors use two different datasets, the authors should describe the sample size and data coverage of each dataset separately. Thank you very much for pointing out the confusion in our data description. We appreciate your detailed and constructive comments. In response, we have made the following clarifications and revisions:

First, regarding the inconsistency in sample sizes, we acknowledge that our earlier description was unclear. For the CHNS-based estimations, the total number of observations used in our analysis is 27,561, covering 12,364 individuals from 4,632 households. Since CHNS is a longitudinal survey, the number of individuals is smaller than the total number of observations due to repeated measurements over time. For the CHIP-based estimations, the total number of observations is 22,199.

Second, we recognize that our original description inaccurately combined details from both datasets, which have different geographical coverage and time spans. CHNS covers selected provinces and provides panel data from 1991 to 2006, while CHIP includes broader geographic coverage and is based on pooled cross-sectional data from 2002, 2007, 2008, and 2013.

To correct this, we have revised the relevant paragraph on page 19. The original statement:

“We combine data from the following sources to generate a 6-year panel across 8 provinces, 40 cities (21 cities are TCZ cities and 19 are non-TCZ cities), 5,058 households and 19,538 individuals to accomplish our research targets.” has been replaced with: “In this research, we combine data from the following sources to accomplish our research targets.”

Details of the data coverage are moved to separate descriptions for each data source. More specifically, in the section introducing the CHNS data (page 21 of the revised manuscript), we add:

“The CHNS sample used in our analysis includes approximately 27,561 observations, covering 8 provinces and 40 cities (21 of which are TCZ cities and 19 are non-TCZ cities). It involves 4,632 households and 12,364 individuals, with data spanning from 1991 to 2006”.

On page 21 of the revised manuscript, which introduces the CHIP data, we add:

“The CHIP sample used in our analysis includes 22,199 observations, covering most regions across the country, with survey waves conducted in 2002, 2007, 2008, and 2013.”

We hope these clarifications address your concerns and improve the clarity and accuracy of our data description. Thank you again for your valuable suggestions.

3) IV regression between SWB, SO2 and wage (Table 4)

Firstly, I would like to confirm this point. In Table 4, only IVs and instrumented endogenous variables are shown. Did the authors use other variables but omit them from the table? The estimation equations (3)-(5) in Section 4.2 include other variables. Or did the authors actually not use any other variables? In other tables, the authors put notes like “Individual control YES”, so it seemed that the authors did not use any other variable in Table 4. I made the following comments, assuming that the authors did not use any other variables. Thank you very much for your insightful comment and for pointing out the ambiguity in our table presentation.

We would like to clarify that the estimations reported in Table 4 were indeed conducted based on Equations (3), (4), and (5), and all control variables included in these equations were used in the regressions. These include individual-level controls (e.g., age, gender, marital status), household-level controls (e.g., household income, household size), and city-level controls (e.g., temperature, precipitation).

However, we acknowledge that we failed to explicitly indicate the inclusion of these control variables in the original version of Table 4, which may have led to the misunderstanding. To address this, we have revised the table by adding appropriate notes: “Individual controls: YES; Household controls: YES; Weather controls: YES; Year FE: YES; City FE: YES; Province-by-year FE: YES” to clarify that these variables were included in the estimation. We have also presented the regression results of all control variables’ coefficients in Table A1 in the Appendix.

We appreciate your careful reading and helpful suggestions, and we believe these changes will improve the clarity and transparency of our presentation.

Then, the use of these IVs can have several problems. Thermal inversion and wind speed themselves can be nice IVs, but they are likely to be correlated to geographical characteristics (thermal inversion is generally more frequent in cold areas and wind speed can be stronger in, say, coastal areas). These characteristics are likely to be correlated to the local economic conditions and people’s SWB. So, to use these IVs, the city-level and province level characteristics, such as weather conditions and development levels, need to be controlled for. Thank you very much for raising this important concern regarding the validity of our instrumental variables.

To address the potential endogeneity arising from the correlation between thermal inversion, wind speed, and underlying geographical or socioeconomic characteristics, we have taken the following steps:

1) Control City Fixed Effects: We include city fixed effects in all IV regressions to account for time-invariant city-level characteristics, such as geography, location (e.g., coastal vs. inland). This helps to mitigate the concern that differences in geography or structural development could bias the IV estimates.

2) Include Province-Year Fixed Effects: To control for time-varying regional shocks and macroeconomic conditions that may influence both air pollution and SWB, we further include province-by-year fixed effects. This controls for province-level trends and annual shocks such as economic policy changes, infrastructure investment, or regional development initiatives.

3) Time-varying city level controls: In addition, we include time-varying weather variables (e.g., temperature, precipitation) at the city level to further mitigate any residual confounding related to short-term weather fluctuations that may affect both pollution levels and well-being.

We believe these strategies jointly address the concern of omitted variable bias and support the validity of our IV approach. Thank you again for your thoughtful and constructive feedback.

The average income as an IV is more questionable. Firstly, living in a high-income area itself can have a direct influence on SWB. Secondly, the average income can cause a residential sorting problem. For example, a high wage area can attract economy-oriented, young and high-skilled individuals who value monetary aspects of SWB greatly. Then the second-stage coefficient of the wage can reflect the effect of such residential sorting, not a pure effect of income on SWB. Thank you very much for your insightful comments regarding the use of provincial-level average industry wage as an instrumental variable for individual income.

We acknowledge the concerns that (1) average income may directly affect SWB, and (2) it may lead to residential sorting, thus violating the exclusion restriction. To address these issues, we have included a range of control variables in our model, including age, education level, marital status, household income, and household size. We apologize for not making this sufficiently clear in the previous version of the table.

In response to your comments, we have further included additional controls such as self-reported health status, city-level population size to better account for potential confounding effects related to regional development and individual health conditions. The updated results, reported in Table A2 of the Appendix, remain consistent with our baseline findings, suggesting that our main conclusions are robust to these concerns.

We greatly appreciate your constructive feedback, which helped us strengthen the empirical validity of our analysis.

Thus, to use these IVs as reliable exogenous factors, I suppose that the authors need to control for potential confounding factors. If the authors actually used other control variables and forgot to mention so, it is fine, but the coefficients of other variables should be provided. If the authors did not use any control variables, then they should add some variables and check if the results remain unchanged. Thank you very much for pointing out the importance of controlling for potential confounding factors when using instrumental variables (IVs).

We would like to clarify that we did include a comprehensive set of control variables in the IV regressions, but we apologize for not stating this clearly in the previous version of the manuscript. Specifically, we controlled for a variety of individual- and household-level characteristics, such as age, education, marital status, household income, and household size, to mitigate possible confounding effects. At the city level, we controlled for city fixed effects to account for time-invariant geographical and policy-related factors, as well as city-specific weather conditions (e.g., temperature and precipitation). In addition, we included province-by-year fixed effects to capture broader macroeconomic and environmental variations across time and regions.

Although these controls were included in the model estimation, we did not clearly report them in the table notes or results. In the revised version, we have now explicitly indicated the inclusion of control variables in the table captions and added the estimated coefficients of these controls in the appendix for transparency and completeness.

We greatly appreciate your careful review and helpful suggestions. We believe these revisions significantly improve the clarity and credibility of our empirical strategy.

4) A related question regarding the IV regression (Table 4)

Why is the sample size in column 1 just 4,224? It must be equal to the second stage result. Did you separately estimate these three equations? Thank you very much for your careful review.

The reason why the sample size in column 1 is different from that in column 3 is because the endogenous variable P_jpt (which is measured by SO2 concentration in city j of province p at time t) itself is a city level observation, while the independent variable 〖SWB〗_ihjkpt in the 2nd stage regression is an individual level measurement associated with individual i from household h. As a result, in the 1st stage regression regarding P_jpt(see Equation (3)), the observation is just 4,224, while in the 2nd stage regression regarding 〖SWB〗_ihjkpt(see Equation(5)), the observation is 22,199. In comparison, since the endogenous variable lnR_ihjkpt is also an individual variable associated with individual i from household h, in the 1st stage regression regarding this variable (see Equation (4) and Column (2)), the observation is 22,199, equals to that in the 2nd stage regression. To match the hierarchical structure of the data, we estimated the 2SLS IV model in two separate steps.

We appreciate your attention to this technical detail and hope this explanation clarifies the issue.

5) Estimation of wages (Table 3)

The authors regarded the column 4 as their main result and used it in the calculation of WTP and the monetary benefit of the TCZ policy. However, I wonder if the column 4 is the best specification.

Firstly, the coefficient of the TCZxPost in column 4 (-26.089) is quite different from those in the other three columns (-15.075 to -16.580). I wonder if the column 4 coefficient is robust.

The difference between column 4 and 3 is the addition of lnGDP. There is no detailed explanation for the definition of this variable, but I suppose that it is the city-level GDP (because the national GDP cannot be used with the year FE, and the province level GDP cannot be used with the province-year FE). Whatever the definition is, however, it is very counterintuitive that this variable has a negative coefficient on wage. It should be positive, at least theoretically. Is there a possibility that lnGDP is causing a multi-collinearity problem and consequently the coefficient of TCZxPost changed greatly?

I actually do not think that lnGDP

---

## [Decision Letter · Decision Letter 2]

22 Jun 2025

Dear Dr. Qiang,

Thank you for submitting your manuscript to PLOS ONE. After careful consideration, we feel that it has merit but does not fully meet PLOS ONE’s publication criteria as it currently stands. Therefore, we invite you to submit a revised version of the manuscript that addresses the points raised during the review process.

We look forward to receiving your revised manuscript.

Kind regards,

Chih-Wei Tseng

Academic Editor

PLOS ONE

Reviewers' comments:

Reviewer's Responses to Questions

**Comments to the Author**

Reviewer #1: (No Response)

Reviewer #2: (No Response)

2. Is the manuscript technically sound, and do the data support the conclusions?

Reviewer #1: Yes

Reviewer #2: Partly

3. Has the statistical analysis been performed appropriately and rigorously?

Reviewer #1: Yes

Reviewer #2: No

4. Have the authors made all data underlying the findings in their manuscript fully available?

Reviewer #1: Yes

Reviewer #2: Yes

5. Is the manuscript presented in an intelligible fashion and written in standard English?

Reviewer #1: Yes

Reviewer #2: Yes

Reviewer #1: I appreciate the efforts made by the authors to address my previous comments. I feel that this version has been well improved. Yet, I still have some questions and suggestions for the authors.

1. I still don’t understand how did you conduct the IV estimation. Did you estimate Eq(3) and Eq(4) separately to predict P and R, or you estimate them all together? Since the set of control variables are different for Eq (3) and Eq (5), it may raise the concern of forbidden regression. Moreover, you should also report the results of weak instruments test and over-identification test to justify the validity of your instruments.

2. Is it possible to control for stricter fixed effect such as individual fixed effects in Eq (5)? The SWB equation is the base of all your calculation, which is very important to support your conclusion. Therefore, I would like to see more robustness checks about the SWB equation.

3. There are inconsistences in the terminology in the paper. For example, In Eq(2), the dependent variable is defined as the average monthly income of individual. However, in Table 3 and 4, the income becomes wage. You should keep the same terminology throughout the paper.

4. To improve the reading experience of the paper, I would suggest to put forward the analysis of SWB before the TCZ analysis. You should first estimate the willingness to pay for air pollution reduction, then proceed to estimate the reduction of air pollution by TCZ policy and calculate monetary value of related improvement of wellbeing.

5. Table 4 is very important, and have you included any control variables? It is not clear in the table. And all tables should have more detailed notes to make it self-readable.

Reviewer #2:

The manuscript was overall adequately revised, and unclear parts were mostly clarified. I also see that the authors have made substantial effort in language editing.

However, I have one major concern which may affect the entire conclusion and credibility of the study. Below, I explain the major concern, followed by minor comments.

**Major concern**

My major concern is the 2SLS estimations. Well, the estimations that the authors call 2SLS but actually seems to be two-step OLS mimicking 2SLS. I initially asked the authors why the sample size in column 1 of Table 4, one of the two first-stages, is just 4,224, while the sample size of the second-stage result is 22,199. The author replied that the column 1 was done at the city-level while others were done at the individual level, implying that each equation was separately estimated. Based on this reply, while the authors call it 2SLS, what the authors actually did seems to be as follows. The second stage has two endogenous regressors (SO2 and wage, denoted by P and R). The authors first estimated the first stages by OLS, using IVs, and obtained the hat-values. Then they plugged the hat-values of the endogenous regressors into the second stage and estimated the second stage again by OLS. Although this is the idea that its name, two-stage least square, implies, the manual implementation like above is a classic mistake because the standard errors are not accurately estimated. The same approach was used in Tables 8 and 11, where the samples were divided by years, and in Tables 13–16, where heterogeneity is explored. Indeed, in the manuscript pp 26–27, these procedures were described, although the authors state “2SLS” but do not mention that they used OLS actually.The problem of this manual procedure mimicking 2SLS by OLS is that the standard errors are not accurately calculated (see e.g., Angrist and Pischke, Mostly Harmless Econometrics, Ch. 4.6 and Wooldridge, Econometric Analysis of Cross Section and Panel Data, Ch. 5.1). Almost always the standard errors are underestimated, making the coefficient of the endogenous regressor overly significant. The authors should use a command for 2SLS, such as ivregress and ivreg2—you are using stata, right? I wonder if the coefficients of endogenous regressors, particularly that of SO2, remain significant after you properly conduct 2SLS with these commands.The authors seem to have a concern that SO2 variable is only at city level and the first stage for SO2 should not be done at individual level, which is the reason that they tried the mimicked 2SLS method. But this is not a concern at all. The authors can simply use ivreg or ivreg2, conducting the first stage for SO2 also at individual level.

In addition, applying this procedure to the heterogeneity analyses (column 3 of Tables 13–16), the authors seem to make a mistake of forbidden regression. I take Table 13 as an example, but the same concerns apply to all other tables. Although the authors do not use the terms “2SLS” and “IV” in the explanations of the equations (11)–(13), the method is actually the same mimicked 2SLS conducted by OLS. There are for endogenous regressors, P, P*developed, R, and R*developed. The authors seemingly used the same P-hat and R-hat as above. And then they interacted P-hat and R-hat with the “developed” dummy variable, and plugged P-hat, R-hat, P-hat*developed and R-hat*developed into the second stage and ran OLS. This is a classic example of forbidden regression.This inappropriate procedure may be the reason that none of the coefficients of the interaction terms, P*developed and R*developed (Table 13) to P*educated and R*educated (Table 16) are significant.Instead, if there are four endogenous regressors, P, P*developed, R, and R*developed, then there must be four first stages, where each endogenous variable is the dependent variable. As the number of IVs are not enough if only Day, Wind, and Avgwage are used as IVs, the authors may use Day*developed, Wind*developed, and R*developed as additional IVs. And these procedures should be done by ivregress, ivreg2 or other appropriate commands, not mimicking 2SLS by OLS.If the interaction terms still do not provide significant coefficients even after appropriate procedure, you may simply drop these heterogeneity analyses.

The other major concerns I raised in the previous review were mostly clarified. However, this major concern is quite a large one. I was not sure at the moment of the previous review whether the explanation in the text is wrong (but the method is accurate) or the method itself has a problem, but it now turns out that the method has the problem of the mimicked-2SLS and forbidden regression.Appropriate re-estimations may affect the significance of the coefficients. In particular, the coefficient of SO2 on SWB in Table 4 is significant at a merely 10% level as of now, and if it remains significant is not clear. This coefficient is of particular importance because it is the main evidence for the inverse logic of the environmental explanation of Easterlin’s Paradox.By the way, just in case the appropriate 2SLS does not provide a significant effect of SO2, I am not sure why the endogeneity of SO2 level is a concern in the first place. The authors only note that “*Since both air quality and income might be correlated with factors which also influence SWB, we apply a series of instrumental variables to deal with potential endogeneity issues* (p25)” and do not argue what kind of unobservable factors confounds the relationship. But is there really any major factor that still affects both SWB and SO2 even after controlling for individual and household characteristics, city FE, and region-year FE? So, one possibility, if the appropriate 2SLS does not provide a significant effect, would be to treat SO2 as exogenous regressor, assuming that potential confounding factors are controlled by FEs. Indeed, Levinson (2012) uses IV only for income and treats pollution as exogenous. I do not know if this approach provides a desired result, but it is better than two-step OLS mimicking 2SLS.

Minor comments

The sample size must be provided in Tables 7, 8, 10, 11, 13–16. In addition, the author should explain what variables are used but omitted in these tables. As for Table 7, I commented before that the demonstration of YES rows and sample sizes were confusing, but I did not recommend you to completely remove them. You may attach notes below the table explaining the sample sizes and the control variables that were used but omitted. You explained these things in text, but it should also be clarified in tables.Table 6 (effects of TCZ on health) and Table 7 (effects of health on SWB) are currently demonstrated in the subsection “Robustness Checks”, but they are not robustness checks. So, you may set up an additional subsection entitled “Mechanisms” or “Health Channel” or whatever and demonstrate them after Table 12—this is not mandatory, however.Regarding the heterogeneity analyses, despite the insignificant differences between groups, the authors describe as if there are differences. For example, regarding the heterogeneity with respect to initial pollution (Table 14), while the authors state that “*the effects of the policy shock are not significantly different across high- and low-pollution cities* ,” they also state that “*Residents in high pollution cities exhibit a lower WTP for pollution reduction* ” (both p45). The authors also state in the conclusion that “*in terms of initial pollution level, it is stronger when the affected city has higher initial pollution level (p49)”.* Well, the WTP differs slightly (287 vs. 337), but this difference reflects the insignificant coefficients of interaction terms—or, in other words, the insignificant coefficients of interaction terms were treated as if they were significantly different from zero. Thus, it is unfair to conclude “*stronger when the affected city has higher initial pollution level* ” as in the conclusion. The same applies to other heterogeneity analyses—although the results themselves may change if you appropriately avoid the methodological problems pointed out above.Capitalization: Some table headings are capitalized (e.g., Table 2. The Impact of Regulation on SO2) whereas some others are not (e.g., Table 4. The relationship between income, air quality and SWB). Be consistent.In addition, although I acknowledge that the authors made significant language editing, further language editing may be needed. For example, there is an incomplete sentence in pp11-12, “*With pooled cross-sectional data covering 214 cities in 22 provinces of China over the years 2002-2013. This dataset matches air pollution and…* ”

**Do you want your identity to be public for this peer review?** For information about this choice, including consent withdrawal, please see our Privacy Policy

Reviewer #1: No

Reviewer #2: No

---

## [Author Response · Author response to Decision Letter 3]

5 Dec 2025

(We have uploaded a Word version of the response letter, in which the replies are

presented more clearly)

Response of Authors to the Comments of Reviewer #1

Comments to the Author

GENERAL COMMENTS

I appreciate the efforts made by the authors to address my previous comments. I feel that this version has been well improved. Yet, I still have some questions and suggestions for the authors.

Thank you very much for your encouraging comments and constructive suggestions. We greatly appreciate your recognition of our previous efforts. We have revised the manuscript carefully in response to your remaining comments. We hope you find the revisions satisfactory.

MAJOR COMMENTS

1) I still don’t understand how did you conduct the IV estimation. Did you estimate Eq(3) and Eq(4) separately to predict P and R, or you estimate them all together? Since the set of control variables are different for Eq (3) and Eq (5), it may raise the concern of forbidden regression. Moreover, you should also report the results of weak instruments test and over-identification test to justify the validity of your instruments.

Thank you for your insightful comment on the instrumental variable (IV) estimation approach, we apologize for the ambiguity in our previous submission that led to your confusion. To address your questions directly:

IV estimation method: Initially, we estimated Eq. (3) and Eq. (4) separately to predict Pollution and Wage. However, we recognize that the different sets of control variables across Eq. (3) and Eq. (5) may may lead to issues related to forbidden regressions, mainly because the previous implementation relied on manually generated fitted values. To address this properly, we now use ivreg2, which automatically performs the correct two-stage least squares estimation and computes valid standard errors.

Weak instrument test: We have supplemented the weak instrument test results in Table 4, including the Cragg-Donald Wald F-statistic (value: 165.782) and the Kleibergen-Paap rk Wald F statistic (value: 140.884). Since these F-statistics far exceed the conventional critical values, the test confirms that our instrument is not weak, validating its relevance.

Over-identification test: In the updated model, we adopt a just-identified specification (one instrumental variable for one endogenous variable: only personal wage is treated as endogenous, while air pollution is treated as relatively exogenous). Over-identification tests (such as the Sargan-Hansen test) require more instruments than endogenous variables to assess instrument exogeneity. As our model is just-identified, this test is not applicable. We have clarified this model setting in the manuscript (p. 26-27) to avoid confusion.

Thank you again for your constructive feedback-this revision has significantly enhanced the methodological rigor and transparency of our work.

2) Is it possible to control for stricter fixed effect such as individual fixed effects in Eq (5)? The SWB equation is the base of all your calculation, which is very important to support your conclusion. Therefore, I would like to see more robustness checks about the SWB equation.

Thank you for your valuable suggestion regarding the robustness of the SWB equation (formerly Eq. 5, now Eq. 4). We fully agree that this equation is foundational to our conclusions.

Regarding the feasibility of adding individual fixed effects: We regret that this is not available for Eq. (4), as the estimation relies on CHIP data, which is pooled cross-sectional data (rather than panel data). Individual fixed effects control for time-invariant unobserved heterogeneity of the same individuals over multiple periods, which requires repeated observations of the same individuals across different time waves.

Although individual fixed effects cannot be used due to the pooled cross-sectional nature of the CHIP data, we conducted several robustness checks-such as year-interacted specifications and geographic restrictions-to verify that the estimates from Eq. (4) remain stable.

The results of these robustness checks confirm that our core findings from Eq. (4) remain consistent and reliable. Thank you again for helping us strengthen the rigor of our analysis.

3) There are inconsistences in the terminology in the paper. For example, In Eq(2), the dependent variable is defined as the average monthly income of individual. However, in Table 3 and 4, the income becomes wage. You should keep the same terminology throughout the paper.

Thank you for pointing out the terminology inconsistency-this is a key detail for clarity, and we appreciate your careful review.

We note that the variable in question (covering annual wage, bonus, and other income from the job) is more accurately defined as income (rather than narrow wage). To resolve the inconsistency:

We have revised all instances of wage in Tables 3, 4, and the corresponding text to R (consistent with the definition in Eq. 2: average monthly income of individual).

We have double-checked the entire manuscript to ensure this terminology is applied uniformly across equations, tables, and descriptions.

This revision aligns the terminology with the actual scope of the variable and eliminates confusion. Thank you again for your meticulous feedback.

4) To improve the reading experience of the paper, I would suggest to put forward the analysis of SWB before the TCZ analysis. You should first estimate the willingness to pay for air pollution reduction, then proceed to estimate the reduction of air pollution by TCZ policy and calculate monetary value of related improvement of wellbeing.

Thank you for your thoughtful suggestion regarding the manuscript’s structure. We truly value your input on improving the reading experience.

After carefully considering your suggestion to present the SWB analysis before the TCZ analysis, we respectfully propose to maintain the current structure for the following reasons, which we believe best serve the logical flow of the research:

The core objective of our study is to examine the causal mechanism of how the TCZ policy (an environmental intervention) affects subjective well-being (SWB). The current organization aligns strictly with this causal chain:

1) Policy Evaluation: We first verify that the TCZ policy effectively reduces air pollution and income (TCZ analysis).

2) Impact Mechanism: We then estimate how this reduction in air pollution and income impacts SWB (SWB equation).

3) Valuation: finally, based on these links, we calculate the Willingness to Pay (WTP) for pollution reduction to quantify the welfare effect.

This order is designed to guide readers naturally from the policy intervention → environmental outcome

→ well-being impact → economic valuation. Placing the SWB analysis first might disrupt this narrative of evaluating a specific policy's welfare effect.

We hope this explanation clarifies the rationale behind our structural choice, and we appreciate your understanding. Thank you again for your valuable feedback.

5) Table 4 is very important, and have you included any control variables? It is not clear in the table. And all tables should have more detailed notes to make it self-readable.

Thank you for your constructive suggestion to improve the clarity of the tables, we fully agree that detailed notes and transparent variable reporting are critical for readability. In response to your concern:

For Table 4: We have revised the table to explicitly include the regression results of control variables (which were previously omitted for brevity) and supplemented the note to clarify this. The updated note for Table 4 now reads: Table 4 reports the estimation results of Equation (3) and Equation (4). Column (1) presents the first-stage IV regression corresponding to Equation (3). The Cragg-Donald Wald F statistic (165.782) and the Kleibergen-Paap rk Wald F statistic (140.884) both exceed conventional thresholds, indicating that the instrument is strongly identified. Column (2) reports the second-stage IV estimates corresponding to Equation (4). (We have also adjusted the table layout to display these control variable coefficients.)

For all tables: We have added detailed self-explanatory notes to each table, which clarify:

1) The corresponding equation for each table;

2) The specification of each column (e.g., time function, fixed effects);

3) Key statistical details (e.g., robust standard errors, significance levels);

4) Supplementary information (e.g., WTP calculation basis in Table 8).

These revisions ensure that each table is self-contained and easy to interpret without relying on the main text. We have checked all tables to confirm the notes align with the results. Thank you again for your meticulous feedback.

Response of Authors to the Comments of Reviewer #2

Comments to the Author

GENERAL COMMENTS

The manuscript was overall adequately revised, and unclear parts were mostly clarified. I also see that the authors have made substantial effort in language editing.

However, I have one major concern which may affect the entire conclusion and credibility of the study. Below, I explain the major concern, followed by minor comments. Thank you very much for your positive overall evaluation of our revised manuscript and for acknowledging our efforts in clarification and language editing. We greatly appreciate the time and care you have devoted to reviewing our work.

We also recognize the major concern you raised regarding the 2SLS estimation procedure. We take this issue very seriously. Below, we address your concern point-by-point and detail the corresponding revisions made to the manuscript.

Major concern

1) My major concern is the 2SLS estimations. Well, the estimations that the authors call 2SLS but actually seems to be two-step OLS mimicking 2SLS. I initially asked the authors why the sample size in column 1 of Table 4, one of the two first-stages, is just 4,224, while the sample size of the second-stage result is 22,199. The author replied that the column 1 was done at the city-level while others were done at the individual level, implying that each equation was separately estimated. Based on this reply, while the authors call it 2SLS, what the authors actually did seems to be as follows. The second stage has two endogenous regressors (SO2 and wage, denoted by P and R). The authors first estimated the first stages by OLS, using IVs, and obtained the hat-values. Then they plugged the hat-values of the endogenous regressors into the second stage and estimated the second stage again by OLS. Although this is the idea that its name, two-stage least square, implies, the manual implementation like above is a classic mistake because the standard errors are not accurately estimated. The same approach was used in Tables 8 and 11, where the samples were divided by years, and in Tables 13–16, where heterogeneity is explored. Indeed, in the manuscript pp 26–27, these procedures were described, although the authors state “2SLS” but do not mention that they used OLS actually.

2) The problem of this manual procedure mimicking 2SLS by OLS is that the standard errors are not accurately calculated (see e.g., Angrist and Pischke, Mostly Harmless Econometrics, Ch. 4.6 and Wooldridge, Econometric Analysis of Cross Section and Panel Data, Ch. 5.1). Almost always the standard errors are underestimated, making the coefficient of the endogenous regressor overly significant. The authors should use a command for 2SLS, such as ivregress and ivreg2—you are using stata, right? I wonder if the coefficients of endogenous regressors, particularly that of SO2, remain significant after you properly conduct 2SLS with these commands.

3) The authors seem to have a concern that SO2 variable is only at city level and the first stage for SO2 should not be done at individual level, which is the reason that they tried the mimicked 2SLS method. But this is not a concern at all. The authors can simply use ivreg or ivreg2, conducting the first stage for SO2 also at individual level.

4) In addition, applying this procedure to the heterogeneity analyses (column 3 of Tables 13-16), the authors seem to make a mistake of forbidden regression. I take Table 13 as an example, but the same concerns apply to all other tables. Although the authors do not use the terms “2SLS” and “IV” in the explanations of the equations (11)-(13), the method is actually the same mimicked 2SLS conducted by OLS. There are for endogenous regressors, P, P*developed, R, and R*developed. The authors seemingly used the same P-hat and R-hat as above. And then they interacted P-hat and R-hat with the “developed” dummy variable, and plugged P-hat, R-hat, P-hat*developed and R-hat*developed into the second stage and ran OLS. This is a classic example of forbidden regression.

5) This inappropriate procedure may be the reason that none of the coefficients of the interaction terms, P*developed and R*developed (Table 13) to P*educated and R*educated (Table 16) are significant.

6) Instead, if there are four endogenous regressors, P, P*developed, R, and R*developed, then there must be four first stages, where each endogenous variable is the dependent variable. As the number of IVs are not enough if only Day, Wind, and Avgwage are used as IVs, the authors may use Day*developed, Wind*developed, and R*developed as additional IVs. And these procedures should be done by ivregress, ivreg2 or other appropriate commands, not mimicking 2SLS by OLS.

7) If the interaction terms still do not provide significant coefficients even after appropriate procedure, you may simply drop these heterogeneity analyses.

8) The other major concerns I raised in the previous review were mostly clarified. However, this major concern is quite a large one. I was not sure at the moment of the previous review whether the explanation in the text is wrong (but the method is accurate) or the method itself has a problem, but it now turns out that the method has the problem of the mimicked-2SLS and forbidden regression.

9) Appropriate re-estimations may affect the significance of the coefficients. In particular, the coefficient of SO2 on SWB in Table 4 is significant at a merely 10% level as of now, and if it remains significant is not clear. This coefficient is of particular importance because it is the main evidence for the inverse logic of the environmental explanation of Easterlin’s Paradox.

10) By the way, just in case the appropriate 2SLS does not provide a significant effect of SO2, I am not sure why the endogeneity of SO2 level is a concern in the first place. The authors only note that “Since both air quality and income might be correlated with factors which also influence SWB, we apply a series of instrumental variables to deal with potential endogeneity issues (p25)” and do not argue what kind of unobservable factors confounds the relationship. But is there really any major factor that still affects both SWB and SO2 even after controlling for individual and household characteristics, city FE, and region-year FE? So, one possibility, if the appropriate 2SLS does not provide a significant effect, would be to treat SO2 as exogenous regressor, assuming that potential confounding factors are controlled by FEs. Indeed, Levinson (2012) uses IV only for income and treats pollution as exogenous. I do not know if this approach provides a desired result, but it is better than two-step OLS mimicking 2SLS.

Thank you very much for your careful and constructive comments on our IV strategy. We truly appreciate the precision with which you identified the issues in our previous implementation. Following your suggestions, we have undertaken a full revision of all empirical estimations involving instrumental variables. Below, we provide a point-by-point response.

1) Correction of the 2SLS implementation

We sincerely appreciate you pointing out the methodological flaw in our previous manual “two-step OLS” procedure. As advised, we have re-estimated all specifications that involve instrumental variables (Tables 4, 8, 10 and the heterogeneity tables) using ivreg2 in Stata, which correctly estimates the first and second stages jointly and provides valid standard errors.

All results previously based on manual 2-step OLS have been removed and replace

---

## [Decision Letter · Decision Letter 3]

19 Dec 2025

Dear Dr. Qiang,

We look forward to receiving your revised manuscript.

Kind regards,

Chih-Wei Tseng

Academic Editor

PLOS One

Journal Requirements:

Reviewers' comments:

Reviewer's Responses to Questions

**Comments to the Author**

Reviewer #1: All comments have been addressed

Reviewer #2: All comments have been addressed

2. Is the manuscript technically sound, and do the data support the conclusions?

Reviewer #1: Yes

Reviewer #2: Partly

3. Has the statistical analysis been performed appropriately and rigorously?

Reviewer #1: Yes

Reviewer #2: Yes

4. Have the authors made all data underlying the findings in their manuscript fully available?

Reviewer #1: Yes

Reviewer #2: Yes

5. Is the manuscript presented in an intelligible fashion and written in standard English?

Reviewer #1: Yes

Reviewer #2: Yes

Reviewer #1: The paper is well revised and the authors have addressed all my comments, I think the paper is ready for publication.

Reviewer #2: The authors addressed the methodological concerns. The overall methodologies are fair.

I suggested in the previous review that the authors may quit instrumenting the SO2 level. The authors previously instrumented it (but in an inappropriate method mimicking 2SLS). The authors admitted that, after using a correct 2SLS method, the significance of the coefficient of SO2 on SWB disappeared, and they chose not to instrument SO2. Certainly, a method that appropriately account for the endogeneity of air pollution is preferrable. However, I at least believe that a method treating SO2 level as exogenous is better than an inappropriate method that wrongly provides a seemingly good result.

The authors also improved the interpretation of their results and removed overstating claims, which I positively evaluate.

Meanwhile, the authors also made various other methodological changes that neither I nor the other reviewer suggested. Making such changes itself is fine if it improves the quality of the study. However, the appropriateness of some of these changes is questionable at times. Below are the main concerns.

1. What is the reason that the sample size changed from the previous version? The observation size for city-level estimations increased from 680 to 745 (Table 2), that in the CHNS based estimations decreased from 27,561 to 11,088 (Table 3) and that in the CHIP based estimations slightly increased from 22,199 to 23,187. The change of the CHNS based estimations is particularly drastic. What is the reason? Did you change the sample criteria? Or, considering the errors having occurred in the previous manuscript, are the sample sizes correct?

2. The addition of P*f(t) in Tables 2, 3 and 6 is questionable. P is the pre-TCZ SO2 emission. This is particularly questionable in Table 2 in which the dependent variable is the SO2 level. The authors previously did not add such a term and used only year FE, city FE, and province-by-year FE, which worked. What is the reason that the authors made this change? The authors explain that they added this term to “control for differential pre-policy trends associated with initial pollution levels (p24)” but it was not necessary in the previous version. Even if the authors try this method, the authors should also show the results in which P*f(t) is not added. In addition, the authors show the parallel trends in Figures 3 and 4, meaning that the differential pre-policy trends are not really a concern. Therefore, the authors should try a model without P*f(t) first and then may try models with P*f(t) as a robustness check.

3. Why did the authors include individual FE for balancing test? It is obvious that the differences in individual and household characteristics disappears once you add individual FE, because it basically absorbs any time-invariant individual-level characteristics. And if you are using a panel dataset in a FE model, the sample balance is not a concern. This table is simply unnecessary.

**Do you want your identity to be public for this peer review?** For information about this choice, including consent withdrawal, please see our Privacy Policy

Reviewer #1: No

Reviewer #2: No

---

## [Author Response · Author response to Decision Letter 4]

22 Jan 2026

Response of Authors to the Comments of Reviewer #1

GENERAL COMMENTS

The paper is well revised and the authors have addressed all my comments, I think the paper is ready for publication.

We sincerely thank the reviewer for the positive evaluation of our manuscript. We greatly appreciate the time and thoughtful comments provided throughout the review process.

Response of Authors to the Comments of Reviewer #2

GENERAL COMMENTS

The authors addressed the methodological concerns. The overall methodologies are fair.

I suggested in the previous review that the authors may quit instrumenting the SO2 level. The authors previously instrumented it (but in an inappropriate method mimicking 2SLS). The authors admitted that, after using a correct 2SLS method, the significance of the coefficient of SO2 on SWB disappeared, and they chose not to instrument SO2. Certainly, a method that appropriately account for the endogeneity of air pollution is preferrable. However, I at least believe that a method treating SO2 level as exogenous is better than an inappropriate method that wrongly provides a seemingly good result.

The authors also improved the interpretation of their results and removed overstating claims, which I positively evaluate.

Meanwhile, the authors also made various other methodological changes that neither I nor the other reviewer suggested. Making such changes itself is fine if it improves the quality of the study. However, the appropriateness of some of these changes is questionable at times. Below are the main concerns.

We sincerely thank the reviewer for the careful evaluation of our revised manuscript and for the constructive feedback provided in the previous and current rounds of review.

Regarding the treatment of SO₂, we agree with your assessment that an appropriately specified model treating SO₂ as exogenous is preferable to an incorrect IV approach, and we have revised the analysis accordingly.

We also acknowledge your concern regarding the additional methodological changes introduced in this revision. These changes were intended to improve the analysis, and we address each of the specific points you raise below.

Major concern

1) What is the reason that the sample size changed from the previous version? The observation size for city-level estimations increased from 680 to 745 (Table 2), that in the CHNS based estimations decreased from 27,561 to 11,088 (Table 3) and that in the CHIP based estimations slightly increased from 22,199 to 23,187. The change of the CHNS based estimations is particularly drastic. What is the reason? Did you change the sample criteria? Or, considering the errors having occurred in the previous manuscript, are the sample sizes correct?

We sincerely thank the reviewer for raising this important question regarding the changes in sample sizes across revisions. Below, we clarify the reasons for each change and confirm that the current samples are correctly constructed and consistent with the revised empirical specifications.

City-level estimations (Table 2: 680 → 745).

In the revised version, we expanded the city-level sample by incorporating additional cities from Heilongjiang Province that were inadvertently omitted in the previous data-matching procedure. This correction improves regional coverage and results in a more complete and representative city-level dataset.

CHNS-based estimations (Table 3: 27,561 → 11,088).

In the previous version, we imputed income using the employment status to increase sample size (e.g., assigning zero income to non-employed individuals). In the revised analysis, we use only income values directly reported in the CHNS and do not impute income based on employment status, which reduces the sample size but improves measurement accuracy.

CHIP-based estimations (Table 4: 22,199 → 23,187).

In the earlier version, observations with zero income were dropped mechanically due to the logarithmic transformation of income. In the revised version, we retain these observations by adding a small constant (0.0001) to income values prior to taking logarithms, which allows us to preserve zero-income observations without materially affecting the estimation results.

Overall, these changes reflect improved data handling and greater internal consistency between variable construction and the empirical specifications. The reported sample sizes follow directly from the updated data construction and processing procedures described above.

2) The addition of P*f(t) in Tables 2, 3 and 6 is questionable. P is the pre-TCZ SO2 emission. This is particularly questionable in Table 2 in which the dependent variable is the SO2 level. The authors previously did not add such a term and used only year FE, city FE, and province-by-year FE, which worked. What is the reason that the authors made this change? The authors explain that they added this term to “control for differential pre-policy trends associated with initial pollution levels (p24)” but it was not necessary in the previous version. Even if the authors try this method, the authors should also show the results in which P*f(t) is not added. In addition, the authors show the parallel trends in Figures 3 and 4, meaning that the differential pre-policy trends are not really a concern. Therefore, the authors should try a model without P*f(t) first and then may try models with P*f(t) as a robustness check.

We sincerely thank the reviewer for this thoughtful and constructive comment regarding the inclusion of the interaction term P×f(t), where P denotes pre-policy SO₂ emissions.

We agree that in many DID applications, models with only unit and time fixed effects can be sufficient when the parallel trends assumption holds. However, in the case of the TCZ policy, city selection was explicitly based on pre-policy pollution severity, which raises concerns about heterogeneous pre-treatment trends across cities with different initial pollution levels.

In this sense, we follow the most recent DID practices (e.g., Li et al., 2016; Liu et al., 2025), and include P×f(t) to flexibly control for potential trend heterogeneity linked to initial SO₂ levels.

In response to your kind suggestion, we also added the estimation results and parallel trend tests from the specifications without P×f(t) in Appendix A (Tables A1–A2 and Figures A1–A2), and added a footnote in the main text to explicitly inform readers of these alternative specifications. In these results, we find that when P×f(t) is excluded, the pre-treatment trend for the logarithm of income is not parallel, and the estimated effect of the TCZ policy on SO₂ is counterintuitively positive. In contrast, once P×f(t) is included, both problematic phenomena are addressed. We therefore view the inappropriate results in the P×f(t) excluded models as outcomes of unaccounted heterogeneous trends and recommend the results from the P×f(t) included models in the main text as credible policy effects.

We sincerely appreciate the reviewer’s constructive suggestion, which has helped us clarify and strengthen our empirical strategy.

3) Why did the authors include individual FE for balancing test? It is obvious that the differences in individual and household characteristics disappears once you add individual FE, because it basically absorbs any time-invariant individual-level characteristics. And if you are using a panel dataset in a FE model, the sample balance is not a concern. This table is simply unnecessary.

We sincerely thank the reviewer for this insightful comment.

The purpose of this balancing test is not to assess sample balance in the conventional sense. Rather, following Tanaka (2015) and Wang et al. (2024), it serves as a supplementary diagnostic to examine whether the TCZ policy variable is systematically correlated with observable time-varying individual-, household-, and city-level characteristics, conditional on individual fixed effects and time controls. This provides additional descriptive evidence on the plausibility of “local randomness” with respect to time-varying covariates.

To streamline the presentation and avoid confusion, we have moved this table to the Appendix and clearly describe it as supplementary evidence. We are also happy to remove it entirely if the editor deems it unnecessary.

---

## [Decision Letter · Decision Letter 4]

26 Jan 2026

Wealth, health, and happiness: An inverse story of the Easterlin Paradox in China

PONE-D-23-32960R4

Dear Dr. Qiang,

We’re pleased to inform you that your manuscript has been judged scientifically suitable for publication and will be formally accepted for publication once it meets all outstanding technical requirements.

Kind regards,

Chih-Wei Tseng

Academic Editor

PLOS One

Additional Editor Comments (optional):

Reviewers' comments:

Reviewer's Responses to Questions

**Comments to the Author**

Reviewer #2: All comments have been addressed

2. Is the manuscript technically sound, and do the data support the conclusions?

Reviewer #2: Yes

3. Has the statistical analysis been performed appropriately and rigorously?

Reviewer #2: Yes

4. Have the authors made all data underlying the findings in their manuscript fully available?

Reviewer #2: Yes

5. Is the manuscript presented in an intelligible fashion and written in standard English?

Reviewer #2: Yes

Reviewer #2: The authors addressed all questions that I raised in the previous review. The manuscript is ready for publication.

**Do you want your identity to be public for this peer review?** For information about this choice, including consent withdrawal, please see our Privacy Policy

Reviewer #2: No

---

## [Editor Report · Acceptance letter]

PONE-D-23-32960R4

PLOS One

Dear Dr. Qiang,

I'm pleased to inform you that your manuscript has been deemed suitable for publication in PLOS One. Congratulations! Your manuscript is now being handed over to our production team.

Kind regards,

on behalf of

Dr. Chih-Wei Tseng

Academic Editor

PLOS One